# Posterior subthalamic nucleus (PSTh) mediates innate fear-associated hypothermia in mice

Can Liu [1,2], Chia-Ying Lee[3], Greg Asher[3], Liqin Cao[3], Yuka Terakoshi[3], Peng Cao [2,4], Reiko Kobayakawa [5], Ko Kobayakawa [5], Katsuyasu Sakurai[3,6✉] & Qinghua Liu [2,3,4,6✉]

The neural mechanisms of fear-associated thermoregulation remain unclear. Innate fear odor 2-methyl-2-thiazoline (2MT) elicits rapid hypothermia and elevated tail temperature, indicative of vasodilation-induced heat dissipation, in wild-type mice, but not in mice lacking Trpa1–the chemosensor for 2MT. Here we report that $Trpa1^{-/-}$ mice show diminished 2MT-evoked c-fos expression in the posterior subthalamic nucleus (PSTh), external lateral parabrachial subnucleus (PBel) and nucleus of the solitary tract (NTS). Whereas tetanus toxin light chain-mediated inactivation of NTS-projecting PSTh neurons suppress, optogenetic activation of direct PSTh-rostral NTS pathway induces hypothermia and tail vasodilation. Furthermore, selective opto-stimulation of 2MT-activated, PSTh-projecting PBel neurons by capturing activated neuronal ensembles (CANE) causes hypothermia. Conversely, chemogenetic suppression of vGlut2+ neurons in PBel or PSTh, or PSTh-projecting PBel neurons attenuates 2MT-evoked hypothermia and tail vasodilation. These studies identify PSTh as a major thermoregulatory hub that connects PBel to NTS to mediate 2MT-evoked innate fear-associated hypothermia and tail vasodilation.

[1] Peking University–Tsinghua University–NIBS Joint Graduate Program, School of Life Sciences, Tsinghua University, Beijing, China. [2] National Institute of Biological Sciences (NIBS), Beijing, China. [3] International Institute for Integrative Sleep Medicine (WPI-IIIS), University of Tsukuba, Tsukuba, Ibaraki, Japan. [4] Tsinghua Institute of Multidisciplinary Biomedical Research (TIMBR), Tsinghua University, Beijing, China. [5] Department of Functional Neuroscience, Institute of Biomedical Science, Kansai Medical University, Osaka, Japan. [6]These authors jointly supervised this work: Katsuyasu Sakurai, Qinghua Liu. ✉email: sakurai.katsuyasu.gm@u.tsukuba.ac.jp; liuqinghua@nibs.ac.cn

The regulation of body temperature—thermoregulation—is essential to life. While ectothermic (cold-blooded) animals, such as reptiles, take on the temperature of their environment, endothermic (warm-blooded) animals, such as mammals, maintain their body temperature in a very narrow range regardless of the ambient temperature[1–5]. In mammals, the body temperature can also be temporally upregulated or downregulated to promote survival in response to specific environmental and physiological challenges[6,7]. For instance, fever is a common physiological response activated by the immune system to combat infections[8,9]. Some animals undergo hibernation or torpor, which are physiologically inactive states characterized by hypothermia and hypometabolism, to conserve energy and survive when temperature or food source is low[10,11].

The body temperature can also be acutely regulated by emotions, such as fear and anger. Fear elicits a series of stereotypical defensive behaviors and physiological responses to enhance animal survival[12]. For example, freezing is a typical defensive behavior in preys to avoid detection by predators[13–15]. Autonomic activities, such as body temperature, heart rate, and blood pressure, can all be regulated as part of the "fight or flight" response under learned and innate fear conditions[16,17]. Depending on the type of external stimuli and animal's internal state, these physiological responses can be quite variable, including hyperthermia or hypothermia, tachycardia or bradycardia, and hypertension or hypotension[12,18,19]. Learned fear, whereby animals are trained by presenting various sensory cues paired with foot shocks, causes elevated core body temperature accompanied by decreased skin temperature in tail and paw[20]. It has also been reported that innate fear stimuli, such as ferret or fox odor, induce hyperthermia in rats[21,22]. On the other hand, long-term restraint/immobilization, hypoxia, or inescapable stress has been shown to evoke hypothermia in mammals[23–25]. In humans, fear or anxiety has a similar effect on body temperature changes and vasoconstriction/vasodilation, creating rapid chills or hot sensation[26,27]. The term "spine chilling" is frequently used to describe extreme fear in many languages. However, the neural mechanisms that underlie various fear-associated rapid changes in body temperature remain largely unknown.

Innate fear odor 2-methyl-2-thiazoline (2MT), a potent analog of fox odor 2,4,5-trimethyl-3-thiazoline (TMT), elicits highly robust innate fear/defensive behaviors, such as freezing, in naive mice[28,29]. Through forward genetics screening of randomly mutagenized mice, we identified the transient receptor potential A1 (Trpa1) mutant as deficient for 2MT/TMT and snake skin-evoked innate fear/defensive behaviors[28]. Furthermore, we demonstrated that Trpa1, a well-known receptor for pungent/irritant chemicals, functions as a chemosensor for 2MT/TMT and that Trpa1[+] trigeminal ganglion (TG) neurons play a major role in 2MT-induced innate freezing[28]. Interestingly, exposure to 2MT can also evoke rapid reduction in cutaneous and core body temperature accompanying innate fear behaviors in wild-type but not $Trpa1^{-/-}$ mice[30]. Here, we investigated the neural mechanisms of 2MT-evoked hypothermia by comparative c-fos mapping and chemo-/opto-genetics-based gain- and loss-of-function experiments. Our studies identify the posterior subthalamic nucleus (PSTh) as a major thermoregulatory hub that connects external lateral parabrachial subnucleus (PBel) to nucleus of the solitary tract (NTS) to mediate 2MT-evoked hypothermia and tail vasodilation.

## Results

We compared the 2MT-induced thermoregulatory response between wild-type and $Trpa1^{-/-}$ mice using a behavior paradigm as previously described[28,30]. To study the effect of 2MT on the change of body temperature, we used infrared (IR) thermography combined with implanted telemetry transmitter to measure in real-time the cutaneous and core body temperature, respectively (Fig. 1a). Although skin temperature normally reflects core body temperature[31], IR thermography allows us to simultaneously measure the temperature of a large body surface, including the tail and trunk areas[32]. Moreover, IR imaging of the body surface is useful to detect additional thermoregulatory alterations, such as vasodilation-induced heat dissipation.

In the absence of fear odor, both the skin temperature (Ts), defined by the highest temperature in the back, and the core body temperature (Tc) remained stable (Fig. 1c, e). Upon 2MT exposure in wild-type mice, both the skin and core body temperature began to decrease immediately in wild-type mice (Fig. 1c, e), with the lowest temperature reached at ~10 min after 2MT exposure (Fig. 1c, e, maximal $\Delta Ts = -4.02 \pm 0.38 \,°C$; maximal $\Delta Tc = -2.73 \pm 0.18 \,°C$). The average $\Delta Ts$ and $\Delta Tc$ was $-2.78 \pm 0.26 \,°C$ and $-2.10 \pm 0.10 \,°C$, respectively, during 2MT exposure (Fig. 1d, f). Interestingly, we also observed a transient sharp increase in the tail temperature (maximal $\Delta Tt = 6.78 \pm 0.08 \,°C$; average $\Delta Tt = 2.29 \pm 0.28 \,°C$) of wild-type mice after 2MT exposure (Fig. 1b, arrow, g, h), which preceded the lowest point of body temperature (Fig. 1c, e, g). Because the mouse tail plays a crucial role in vasodilation-induced heat dissipation[33], these observations suggest that heat dissipation over the tail surface contributes critically to 2MT-evoked hypothermia. By contrast, we observed no significant changes in the tail, skin, or core body temperature (maximal $\Delta Tt = 0.52 \pm 0.10 \,°C$; maximum $\Delta Ts = -0.54 \pm 0.05 \,°C$; maximum $\Delta Tc = -0.31 \pm 0.07 \,°C$) in $Trpa1^{-/-}$ mice following 2MT treatment (Fig. 1b–h). Moreover, we found that exposure to cinnamaldehyde, a well-known Trpa1 agonist, did not induce hypothermia in wild-type mice (Supplementary Fig. 1a, b). Taken together, these results indicate that innate fear odor 2MT induces rapid hypothermia and elevated tail temperature in wild-type mice, which are both abolished in mice deficient for Trpa1, the chemosensor for 2MT.

**Comparative analysis of 2MT-induced c-fos expression in wild-type and $Trpa1^{-/-}$ mice.** To investigate the neural mechanism responsible for 2MT-induced hypothermia, we performed immunohistochemistry to compare the expression of immediate early gene c-fos in many brain regions of wild-type and $Trpa1^{-/-}$ mice following 2MT exposure (Fig. 1i, j and Supplementary Fig. 1c–i). Consistently, we observed that $Trpa1^{-/-}$ mouse brains exhibited diminished c-fos expression relative to wild-type mouse brains in several known fear/stress, thermoregulation, and vasodilation/vasoconstriction centers, such as the central amygdala (CeA), paraventricular hypothalamic nucleus (PVN), and ventrolateral periaqueductal gray (vlPAG), as well as PSTh, PBel, and NTS (Fig. 1i, j and Supplementary Fig. 1c–e)[4,15,34–37]. Conversely, we found significantly more c-fos expressing neurons in the preoptic area (POA), such as the median preoptic nucleus (MnPO) and ventromedial preoptic nucleus (VMPO) in $Trpa1^{-/-}$ mice relative to wild-type mice (Supplementary Fig. 1g, h). Because neurons that are inhibited during 2MT response will not be c-fos-positive, this observation suggests a possibility that some unknown neural pathway may contribute to 2MT-evoked hypothermia through suppression of MnPO/VMPO neurons. By contrast, we did not observe significant difference in c-fos expression in the dorsolateral periaqueductal gray (dlPAG) and dorsal part of the lateral parabrachial nucleus (LPBD) (Supplementary Fig. 1f, i). The latter observation suggests that the thermoregulatory center LPBD is unlikely to play a critical role in 2MT-evoked hypothermia.

It is well-known that cutaneous warm and cold sensory signals are conveyed from the lateral parabrachial nucleus (LPB) to the

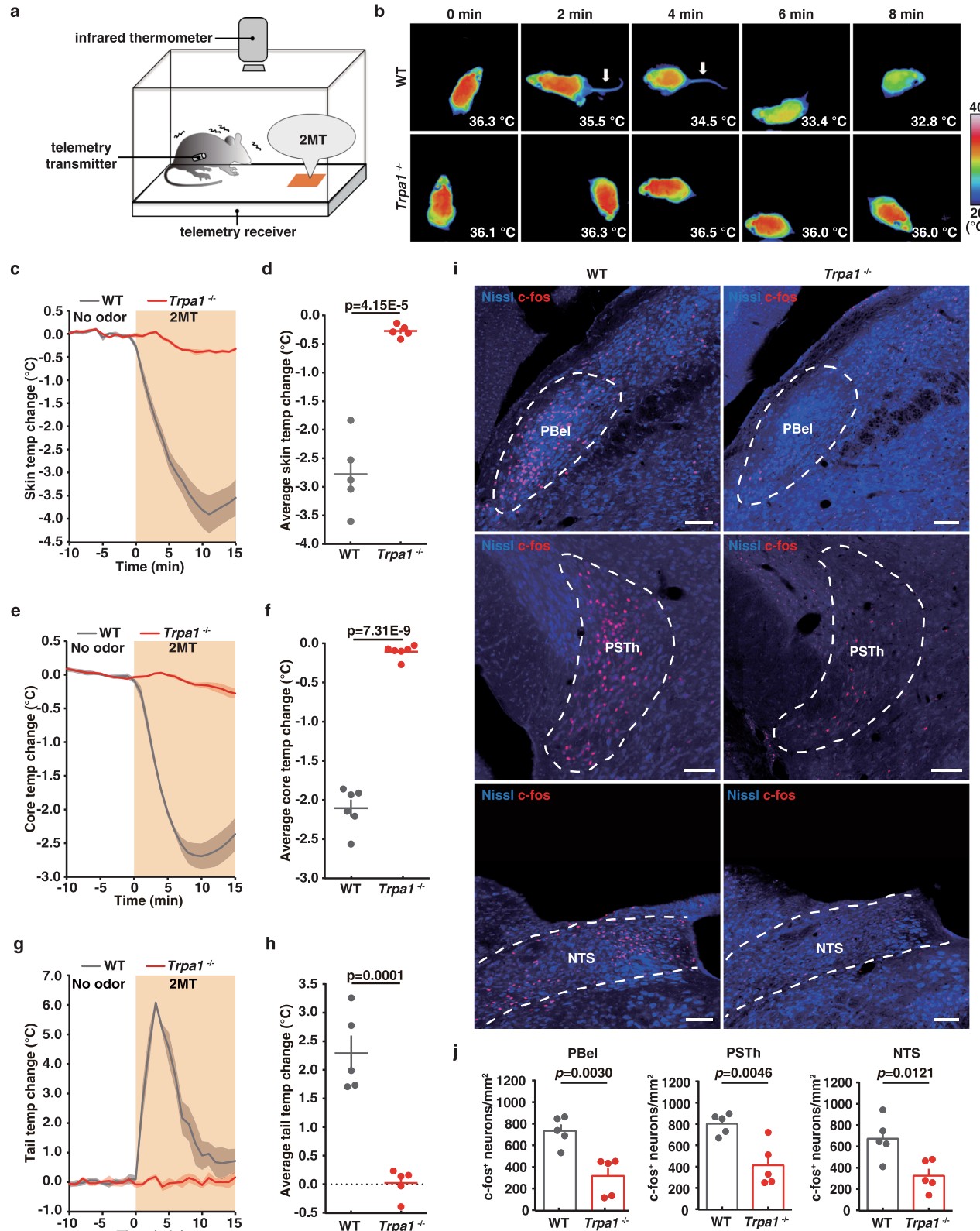

**Fig. 1 Innate fear odor 2MT induces hypothermia and tail temperature increase via Trpa1. a** Schematic of 2MT-induced hypothermia assay in mice. **b** Time-lapsed thermal images showing tail and skin temperature changes in wild-type (upper) and *Trpa1*$^{-/-}$ (bottom) mice during 2MT treatment. Skin (**c**), core (**e**) and tail (**g**) temperature curves of wild-type and *Trpa1*$^{-/-}$ mice (*n* = 5 or 6) before and during 2MT treatment. Average skin (**d**), core (**f**) and tail (**h**) temperature changes of wild-type and *Trpa1*$^{-/-}$ mice (*n* = 5 or 6) during 2MT treatment. **i** Representative images showing 2MT-induced c-Fos expression in PBel, PSTh, and NTS of wild-type and *Trpa1*$^{-/-}$ mice. **j** Quantitative analysis of 2MT-evoked c-Fos expression in PBel, PSTh, and NTS in wild-type and *Trpa1*$^{-/-}$ mice (*n* = 5). **c–h**, **j** Data are mean ± SEM; two-side Student's *t* test. Scale bars, 100 μm.

POA by two distinct glutamatergic pathways, the LPBD and LPBel. PBel contains a functionally heterogeneous neural population involved in the processing of diverse sensory information, such as temperature, taste, and pain[1–3,38–44]. PSTh is well recognized as a motor controlling center[45], of which deep brain stimulation is employed to treat Parkinson's disease (PD)[46,47]. It has recently been reported that optogenetic stimulation of axon terminals of PBel-CGRP[+] (calcitonin gene-related peptide) neurons in PSTh induces a reduction in tail temperature[48]. NTS is the crucial relay nucleus for many thermoregulation-related signal transmission[4,34,35]. For example, adenosine A1 receptor activation in NTS induces hypothermia[34]. Additionally, NTS activity is involved in the brown adipose tissue (BAT)-dependent thermoregulation[4,35]. Based on the differential c-fos expression between wild-type and $Trpa1^{-/-}$ mouse brains following 2MT exposure, we hypothesized that the PBel, NTS, and PSTh may be directly involved in 2MT-evoked hypothermia.

**NTS neurons receive synaptic inputs from PSTh neurons**. A major autonomic response to reduce body temperature is through the vasodilation-induced heat dissipation in the glabrous organs, such as the mouse tail[3]. Accordingly, we observed that 2MT exposure elicited an acute elevation of tail temperature, indicative of vasodilation-induced heat dissipation, prior to reaching the lowest point of body temperature (Fig. 1b–h). It has been reported that NTS activation may induce tail vasodilation through inhibition of neural activity in the rostral ventrolateral medulla (RVLM), a pivotal vasomotor control nucleus in the brain stem[49]. Thus, we hypothesized that the upstream input nuclei of NTS could play an important role in 2MT-induced hypothermia and elevated tail temperature.

To identify the upstream nuclei that project to NTS, we performed retrograde tracing of NTS neurons by unilateral injection of cholera toxin B subunit (CTB)-conjugated Alexa Fluor 488 (CTB-488) into the NTS (Fig. 2a and Supplementary Fig. 2a). CTB-injected mice were exposed to 2MT to visualize 2MT-activated c-fos-expressing neurons by immunohistochemistry. In the NTS-projecting nuclei involved in 2MT-evoked hypothermia, we expected that the CTB-488 labeled neurons would show significant overlap with 2MT-activated c-fos-labeled neurons. Accordingly, we observed CTB-positive neurons in the PSTh, PBel, CeA, and PVN (Fig. 2b and Supplementary Fig. 2b–d). Among these, PSTh exhibited the highest density of CTB-positive neurons and the highest percentage (~32%) of c-fos[+], CTB[+] double-positive neurons (Fig. 2b and Supplementary Fig. 2e–g), implicating a potential involvement of the PSTh-NTS pathway in 2MT-evoked hypothermia response. It should be noted that the highest CTB/c-fos overlap implies a potential direct projection from 2MT-activated neurons in PSTh to NTS. However, indirect pathways from c-fos[+] local interneurons to CTB-labeled projection neurons could also play important roles, and we choose to focus on the direct PSTh-NTS pathway in this study.

**vGlut2[+] PSTh neurons play a major role for 2MT-evoked hypothermia and tail vasodilation**. To determine the neurotransmitter type of 2MT-activated neurons in PSTh, we performed two-color fluorescence in situ hybridization using antisense RNA probes for c-fos and vesicular glutamate transporter 2 (vGlut2) or vesicular GABA transporter (vGAT). Our results indicated that the majority of 2MT-activated c-fos-positive PSTh neurons expressed vGlut2 (87 ± 2%), whereas a minority were vGAT-positive (3 ± 1%) (Fig. 2c). Thus, the majority of 2MT-activated PSTh neurons are vGlut2-expressing excitatory neurons.

To examine whether vGlut2[+] PSTh neurons were critical for 2MT-evoked hypothermia, we utilized the Designer Receptors Exclusively Activated by Designer Drugs (DREADDs) system for specific suppression of vGlut2[+] PSTh neurons. We bilaterally injected AAV2/9-hSyn-DIO-hM4Di-mCherry or AAV2/10-EF1a-DIO-mCherry into the PSTh of vGlut2-IRES-Cre mice to specifically express hM4Di or mCherry in vGlut2[+] PSTh neurons (Fig. 2d, e). To suppress the neural activity of vGlut2[+] PSTh neurons, clozapine analog compound 21 (C21, 1 mg/kg) was intraperitoneally (i.p.) administered in the mCherry- or hM4Di-expressing mice 30 min before 2MT exposure. Chemogenetic inhibition of vGlut2[+] PSTh neurons significantly blunted the 2MT-evoked hypothermia and tail temperature increment in hM4Di-expressing mice compared to mCherry-expressing mice (average ΔTt: hM4Di, 1.09 ± 0.14 °C vs. mCherry, 2.47 ± 0.38 °C, $p = 0.0097$; average ΔTs: hM4Di, −1.37 ± 0.12 °C vs. mCherry, −2.14 ± 0.16 °C, $p = 0.0027$; average ΔTc: hM4Di, −1.44 ± 0.12 °C vs. mCherry, −2.29 ± 0.24 °C; $p = 0.0061$) (Fig. 2f–l). These results suggest that the activity of vGlut2[+] PSTh neurons plays a major role in mediating 2MT-evoked hypothermia in mice.

**Tetanus toxin light chain (TeLC)-mediated inhibition of NTS-projecting PSTh neurons diminished 2MT-evoked hypothermia and tail vasodilation**. To determine whether direct PSTh-NTS pathway was critical for 2MT-evoked hypothermia, we specifically express the TeLC in the NTS-projecting PSTh neurons through bilateral injections of AAV2/retro-hSyn-Cre expressing retrogradely transported Cre recombinase into the NTS, and AAV2/9-hSyn-FLEX-TeLC-P2A-GFP expressing Cre-dependent TeLC into the PSTh, respectively (Fig. 3a, b). For control mice, AAV2/9-hSyn-FLEX-EGFP-expressing Cre-dependent EGFP was bilaterally injected instead into the PSTh to label the NTS-projecting PSTh neurons. We found that TeLC-mediated silencing of neurotransmission in NTS-projecting PSTh neurons abrogated 2MT-evoked tail temperature increase (Fig. 3c–e, average ΔTt: TeLC, 0.34 ± 0.11 °C vs. EGFP, 1.40 ± 0.17 °C, $p = 0.0091$) and blunted 2MT-evoked hypothermia (Fig. 3f–i, average ΔTs: TeLC, −1.12 ± 0.12 °C vs. EGFP, −1.83 ± 0.13 °C, $p = 0.0210$; average ΔTc: TeLC, −0.97 ± 0.21 °C vs. EGFP, −2.00 ± 0.18 °C, $p = 0.0214$). Moreover, we observed an excellent positive correlation between the virus transduction rate and the suppression effect of 2MT-evoked hypothermia: the higher the number of TeLC-labeled NTS-projecting PSTh neurons, the lesser the reduction in the skin and core body temperature following 2MT treatment (Supplementary Fig. 3a, b). These results suggest that 2MT exposure activates the PSTh-NTS pathway to induce hypothermia and tail temperature increment in mice. However, we cannot exclude the possibility of existence and involvement of collateral projection of NTS-projecting PSTh neurons to other brain areas for 2MT-induced hypothermia.

**Opto-stimulation of direct PSTh-RNTS pathway causes hypothermia and tail vasodilation**. To test whether the activation of direct PSTh-NTS pathway could induce hypothermia, we specifically labeled the NTS-projecting PSTh neurons by bilateral injection of AAV2/retro-hSyn-ChR2-EYFP or AAV2/retro-hSyn-EGFP into the NTS for retrograde expression of Channelrhodopsin 2 (ChR2) or EGFP, respectively, and by implanting optical fibers above the PSTh (Fig. 4a, b). Blue light stimulation of the NTS-projecting PSTh neurons resulted in hypothermia accompanied by elevated tail temperature in the ChR2-expressing mice (Fig. 4c). Quantitative analysis indicated that the activation of NTS-projecting PSTh neurons triggered a significant surge in the tail temperature (average ΔTt: ChR2, 3.96 ± 0.39 °C vs. EGFP, −0.03 ± 0.05 °C; $p < 0.0001$), as well as the reduction in the skin and core body temperature (average ΔTs: ChR2, −1.92 ± 0.23 °C

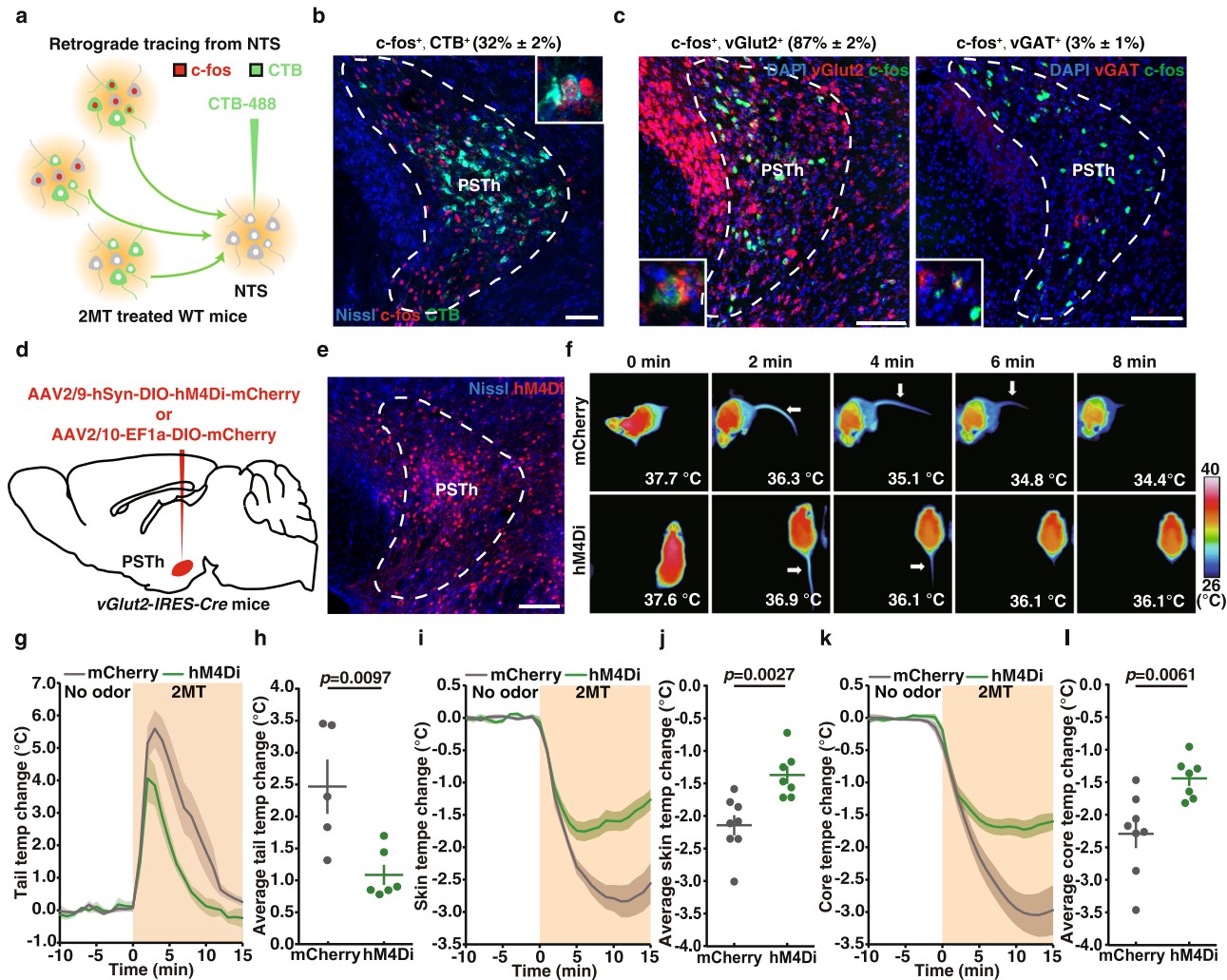

**Fig. 2 Inhibition of vGlut2⁺ PSTh neurons attenuates 2MT-evoked hypothermia and tail temperature increase. a** Schematic of retrograde tracing from NTS neurons by CTB injection. **b** Representative image showing double immunostaining of CTB and c-fos. The percentage of double-positive neurons (CTB⁺, c-fos⁺/CTB⁺%) in PSTh is in parenthesis ($n = 3$). **c** Representative images and quantitative analysis showing the percentage of *c-fos⁺, vGlut2⁺* or *c-fos⁺, vGAT⁺* double-positive neurons among *c-fos⁺* neurons in PSTh by two-color in situ hybridization ($n = 3$). **d** Schematic of chemogenetic inhibition experiment of vGlut2⁺ PSTh neurons in *vGlut2-IRES-Cre* mice. **e** Representative image of hMD4i-labeled vGlut2⁺ PSTh neurons ($n = 7$). **f** Time-lapsed thermal images of mCherry-expressing mice and hM4Di-expressing mice during 2MT treatment following administration of C21. Tail (**g**), skin (**i**), and core (**k**) temperature curves of mice with (hM4Di, $n = 7$) or without (mCherry, $n = 8$) inactivation of vGlut2⁺ PSTh neurons before and during 2MT treatment. Average tail (**h**), skin (**j**), and core (**l**) temperature changes of mice with (hM4Di, $n = 7$) or without (mCherry, $n = 8$) inactivation of vGlut2⁺ PSTh neurons during 2MT treatment. A few mice were not included in the analysis of tail temperature (**e**) because their tails were frequently obscured in the thermal images. **g–l** Data are mean ± SEM; two-side Student's *t* test. Scale bars, 100 µm.

vs. EGFP, $-0.35 \pm 0.05\,°C$, $p < 0.0001$; average ΔTc: ChR2, $-1.81 \pm 0.21\,°C$ vs. EGFP, $-0.31 \pm 0.06\,°C$, $p < 0.0001$) in ChR2-expressing mice, but not in EGFP-expressing control mice (Fig. 4d–i). It should be noted that a potential caveat of this experiment was that retrograde-labeled PSTh neurons could project to other brain areas besides NTS that might also be involved in thermoregulation.

To further confirm whether the PSTh-NTS pathway was involved in thermoregulation, we specifically labeled vGluT2⁺ PSTh neurons by bilateral injection of AAV2/9-EF1a-DIO-ChR2-mCherry or AAV2/9-EF1a-DIO-mCherry into the PSTh of *vGlut2-IRES-Cre* mice for expression of ChR2 or mCherry, and implanted optical fibers on the NTS (Fig. 5a). Because NTS is a rather large area, we implanted optic fibers above either the rostral (RNTS; AP coordinate $-7.0$) or caudal (CNTS; AP coordinate $-7.8$) side of NTS to distinguish which part of NTS was responsible for 2MT-induced hypothermia and tail

temperature increase (Fig. 5a, b). Blue light stimulation of the axon terminals of PSTh neurons in RNTS resulted in robust hypothermia accompanied by elevated tail temperature in the ChR2-expressing mice, but not in mCherry-expressing mice (average ΔTt: ChR2, $4.17 \pm 0.62\,°C$ vs. mCherry, $0.03 \pm 0.26\,°C$; $p = 0.0034$; average ΔTs: ChR2, $-2.39 \pm 0.16\,°C$ vs. mCherry, $-0.78 \pm 0.10\,°C$, $p = 0.0012$; average ΔTc: ChR2, $-1.73 \pm 0.25\,°C$ vs. mCherry, $-0.26 \pm 0.34\,°C$, $p = 0.0308$) (Fig. 5c–j and Supplementary Fig. 4). By contrast, optogenetic stimulation of CNTS-projecting PSTh neurons showed no statistically significant change in either tail temperature or skin/core body temperature relative to the control mCherry-expressing mice (average ΔTt: ChR2, $2.05 \pm 1.02\,°C$ vs. mCherry, $0.06 \pm 0.12\,°C$; $p = 0.2248$; average ΔTs: ChR2, $-1.35 \pm 0.17\,°C$ vs. mCherry, $-0.87 \pm 0.10\,°C$, $p = 0.6652$; average ΔTc: ChR2, $-0.30 \pm 0.09\,°C$ vs. mCherry, $-0.05 \pm 0.06\,°C$, $p = 0.1067$) (Fig. 5c–j and Supplementary Fig. 4). These results suggest that the direct PSTh-RNTS

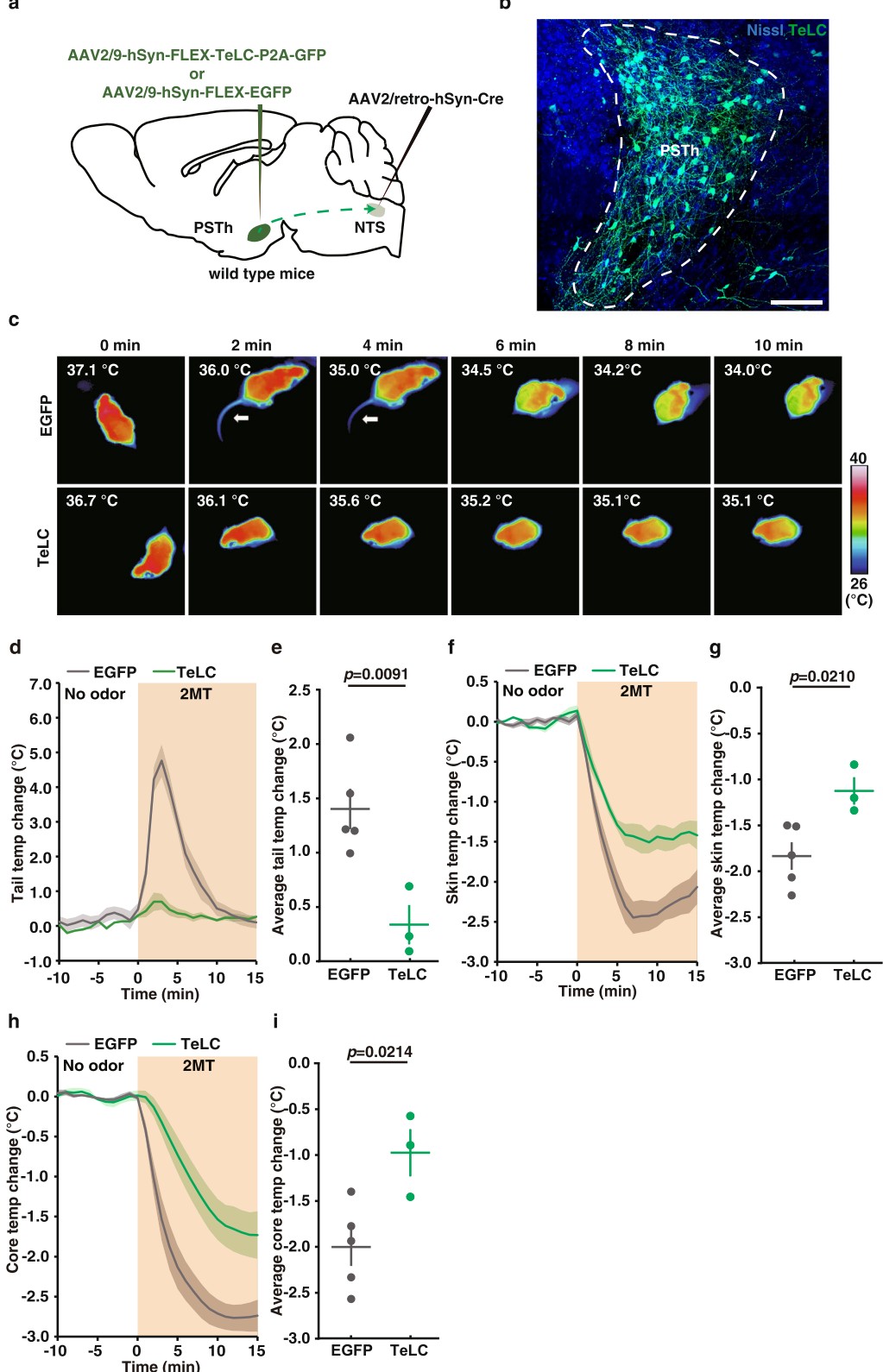

**Fig. 3 TeLC-mediated inactivation of NTS-projecting PSTh neurons diminishes 2MT-evoked hypothermia and abrogates tail temperature increase.**
**a** Schematic of TeLC-mediated inactivation of PSTh neurons in wild-type mice. **b** Representative image showing TeLC-GFP-labeled NTS-projecting PSTh neurons ($n = 5$). **c** Time-lapsed thermal images of wild-type mice without (EGFP, up) or with (TeLC, bottom) inactivation of NTS-projecting PSTh neurons during 2MT treatment. Tail (**d**), skin (**f**), and core (**h**) temperature curves of EYFP-expressing ($n = 5$) and TeLC-expressing ($n = 3$) mice before and during 2MT treatment. Average tail (**e**), skin (**g**), and core (**i**) temperature changes of EGFP-expressing ($n = 5$) and TeLC-expressing ($n = 3$) mice during 2MT treatment. Two TeLC-expressing mice were not included in the analysis because of low virus transduction rate (Supplementary Fig. 3). **d–i** Data are mean ± SEM; two-side Student's $t$ test. Scale bar, 100 μm.

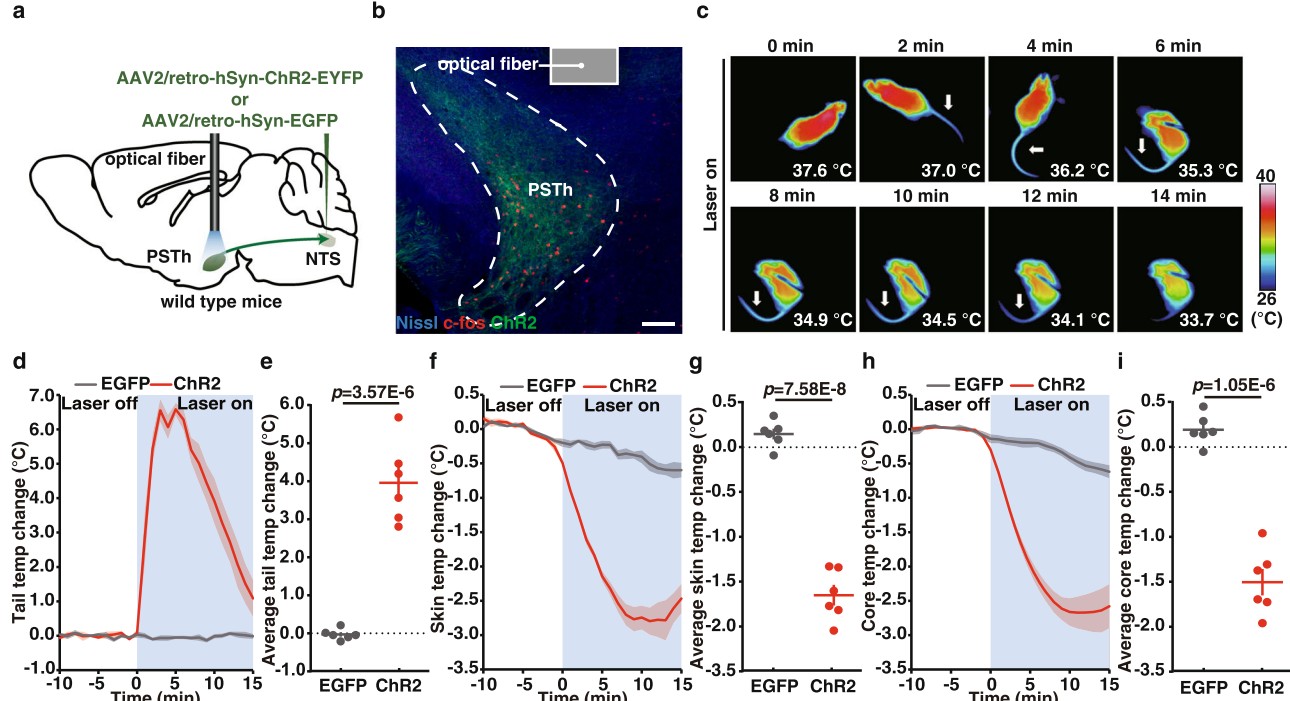

**Fig. 4 Activation of NTS-projecting PSTh neurons evokes hypothermia and tail temperature increase. a** Schematic of optogenetic activation of the PSTh-NTS pathway in wild-type mice. **b** Representative image showing c-fos expression in ChR2-labeled PSTh neurons after photoactivation ($n = 6$). **c** Time-lapsed thermal images of ChR2-expressing mice during photoactivation of the NTS-projecting PSTh neurons. Tail (**d**), skin (**f**), and core (**h**) temperature curves of EGFP-expressing ($n = 6$) and ChR2-expressing mice ($n = 6$) before and during photoactivation of the NTS-projecting PSTh neurons. Average tail (**e**), skin (**g**), and core (**i**) temperature changes of EGFP-expressing ($n = 6$) and ChR2-expressing mice ($n = 6$) during photoactivation of NTS-projecting PSTh neurons. **d–i** Data are mean ± SEM; two-side Student's $t$ test. Scale bar, 100 μm.

pathway plays a major role in mediating 2MT-evoked hypothermia in mice.

**PSTh neurons receive synaptic inputs from PBel neurons**. To study how 2MT activated PSTh neurons to induce hypothermia, we performed retrograde tracing to find upstream input nuclei of PSTh neurons by unilateral injection of CTB-488 into the PSTh (Fig. 6a and Supplementary Fig. 5a). CTB-488 injected mice were exposed to 2MT to identify the CTB$^+$, c-fos$^+$ double-positive neurons, which indicated the PSTh-projecting neurons that were also activated by 2MT exposure. While many CTB$^+$ neurons were found in the PBel, NTS, PVN, and CeA, the PBel showed the highest density of CTB$^+$ neurons and the highest percentage of c-fos$^+$, CTB$^+$ double-positive neurons (Fig. 6b and Supplementary Fig. 5b–g). Consistent with a recent report[36,48], our results suggest that PSTh neurons receive synaptic input from PBel neurons.

**Opto-stimulation of 2MT-activated, PSTh-projecting PBel neurons causes hypothermia**. If the PSTh-projecting PBel neurons are causally linked to 2MT-evoked hypothermia, then specific activation of these neurons may be sufficient to induce similar phenotypes to that caused by 2MT exposure. To test this possibility, we used the capturing activated neuronal ensembles (CANE) system to selectively label and manipulate the neurons expressing c-fos in response to 2MT exposure[41,50,51]. Briefly, singly housed Fos$^{TVA}$ male mice were exposed to 2MT to induce transient c-fos and TVA expression in PBel neurons. After 1 h, 2MT-treated mice were bilaterally injected in the PBel with AAV2/9-EF1a-DIO-ChR2-mCherry and CANE-Lv-Cre, a pseudotyped lentivirus that could specifically infect the TVA-expressing PBel neurons and express Cre recombinase (Fig. 6c, d). Thus, the use of CANE system restricted the expression of

ChR2 to only 2MT-activated PBel neurons. In control mice, 2MT-activated PBel neurons were selectively labeled with mCherry by bilateral injection of AAV2/10-EF1a-DIO-mCherry and CANE-Lv-Cre in the PBel following 2MT exposure (Fig. 6c).

Next, we implanted optic fibers above the virus-injected PBel region for optogenetic stimulation of the cell bodies of 2MT-activated PBel neurons (Fig. 6c, d). We observed a significant reduction in the skin and core body temperature (average ΔTs: ChR2, $-1.20 \pm 0.21$ °C vs. mCherry, $-0.40 \pm 0.14$ °C; $p = 0.0147$; average ΔTc: ChR2 $-1.16 \pm 0.18$ °C vs. mCherry, $-0.13 \pm 0.20$ °C; $p = 0.0054$), accompanied by a transient increase of the tail temperature (average ΔTt: ChR2, $1.39 \pm 0.40$ °C vs. mCherry, $0.14 \pm 0.04$ °C; $p = 0.0296$) in the ChR2-expressing, but not mCherry-expressing, Fos$^{TVA}$ mice during blue light stimulation (Fig. 6e–k). These results indicate that selective stimulation of 2MT-activated PBel neurons effectively caused a hypothermic response similar to that caused by 2MT exposure in mice.

Consistent with the retrograde tracing results of PSTh neurons (Fig. 6a, b), we found that the axonal projections of 2MT-activated PBel neurons terminated in the PSTh (Fig. 6m). To test whether the PBel-PSTh pathway was involved in thermoregulation, we implanted optic fibers above the PSTh and stimulated the axon terminals of 2MT-activated PBel neurons by blue light (Fig. 6l, m). Optogenetic stimulation of the axon terminals in PSTh induced a significant reduction in the skin and core body temperature (average ΔTs: ChR2, $-1.07 \pm 0.20$ °C vs. mCherry, $-0.29 \pm 0.15$ °C, $p = 0.0231$; average ΔTc: ChR2, $-1.28 \pm 0.18$ °C vs. mCherry, $-0.01 \pm 0.21$ °C, $p = 0.0017$) in ChR2-expressing mice, but not in mCherry-expressing mice (Fig. 6n–q). However, we did not observe a significant change in the tail temperature in all of the mice during blue light stimulation (data not shown), which was probably due to the relatively low efficiency of neuron

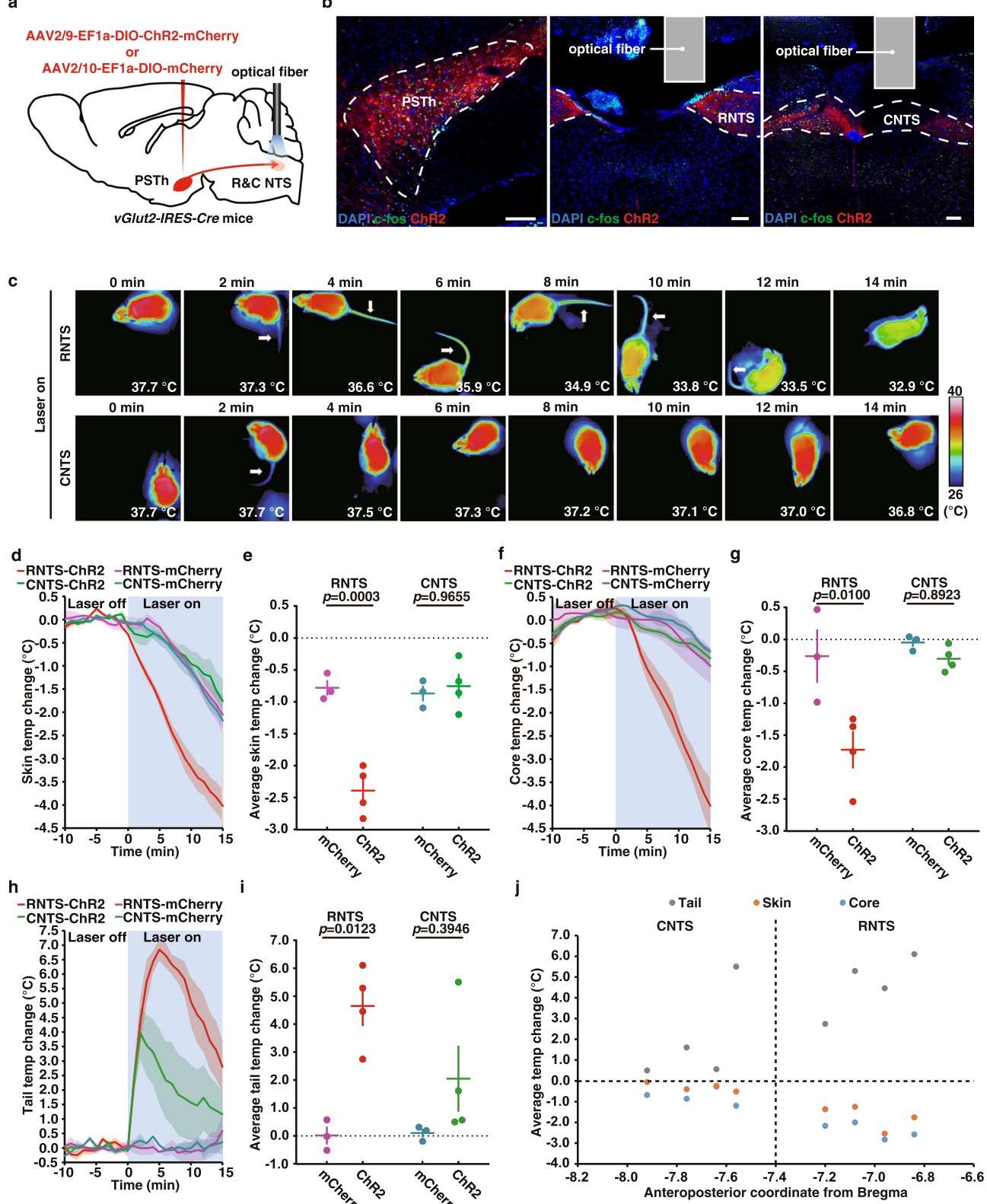

labeling by CANE and/or inability to activate the sufficient number of PBel-PSTh axonal terminals.

An alternative explanation is that the 2MT-activated PBel neurons may also send projections to other thermoregulatory regions to induce the hypothermic responses. It has been reported that a group of PBel neurons respond to cold temperature and project to the POA—a well-known thermoregulatory center[42–44,52,53]. However, we found that 2MT-activated PBel neurons did not project to the POA (Supplementary Fig. 6a), suggesting that they represent a separate neural population from those responding to cold temperature and project to POA.

Interestingly, we observed that the CANE-labeled PBel neurons also projected to the CeA, a well-known region associated with fear responses (Supplementary Fig. 6b). Thus, we optogenetically

**Fig. 5 RNTS, but not CNTS, is the main target for hypothermia evoking PSTh neurons. a** Schematic of optogenetic activation of the PSTh-NTS pathway in wild-type mice. **b** Representative images showing c-fos expression in ChR2-labeled PSTh neurons (left) and their axon terminals in RNTS (middle) and CNTS (right) in *vGlut2-IRES-Cre* mice after photoactivation. We stained $n = 8$ ChR2-expressing mice (four for RNTS, four for CNTS) and obtained similar results. **c** Time-lapsed thermal images of ChR2-expressing mice during photoactivation of axon terminals of RNTS- or CNTS-projecting PSTh neurons. Skin (**d**), core (**f**), and tail (**h**) temperature curves of mCherry-expressing ($n = 3$) and ChR2-expressing mice ($n = 4$) before and during photoactivation of axon terminals of RNTS- or CNTS-projecting PSTh neurons. Average skin (**e**), core (**g**), and tail (**i**) temperature changes of mCherry-expressing ($n = 3$) and ChR2-expressing ($n = 4$) mice during photoactivation of axon terminals of RNTS- or CNTS-projecting PSTh neurons. **j** Correlative analysis between the photoactivation-induced average tail, skin, and core temperature changes in ChR2-expressing mice and the anteroposterior coordinates of the optical fiber implant sites. **d–i** Data are mean ± SEM; two-way ANOVA analysis followed by Tukey's multiple comparisons test. Scale bars, 100 μm.

stimulated the axon terminals of 2MT-activated PBel neurons in the CeA by blue light, which resulted in a modest reduction in skin and core body temperature (Supplementary Fig. 6c–g, average ΔTs: ChR2, $-0.87 \pm 0.15$ °C vs. mCherry, $-0.31 \pm 0.04$ °C, $p = 0.0577$; average ΔTc: ChR2, $-0.68 \pm 0.12$ °C vs. mCherry, $-0.17 \pm 0.11$ °C, $p = 0.0320$), but no change in the tail temperature (data not shown). Thus, in addition to PSTh, other brain areas downstream of PBel neurons, such as CeA, may also contribute to 2MT-induced hypothermic responses.

**The PBel-PSTh direct pathway plays a major role for 2MT-evoked hypothermia and tail vasodilation.** To test whether the activity of PBel neurons was involved in 2MT-evoked hypothermia and tail temperature increase, we carried out a cell type-specific DREADDs inhibition experiment. First, we showed by two-color in situ hybridization that the majority (~86%) of 2MT-activated PBel neurons are vGlut2-expressing excitatory neurons (Supplementary Fig. 7a). Second, we bilaterally injected AAV2/9-hSyn-DIO-hM4Di-mCherry or AAV2/10-EF1a-DIO-mCherry into the PBel of *vGlut2-IRES-Cre* mice to specifically express hM4Di or mCherry in the vGlut2$^+$ PBel neurons (Supplementary Fig. 7b, c). Administration of C21 blunted 2MT-induced tail temperature increase (average ΔTt: hM4Di, $0.56 \pm 0.26$ °C vs. mCherry, $1.71 \pm 0.23$ °C, $p = 0.0130$) in the hM4Di-expressing mice relative to mCherry-expressing mice (Supplementary Fig. 7d–f). Additionally, chemogenetic suppression of vGlut2$^+$ PBel neurons also slightly attenuated 2MT-evoked hypothermia (average ΔTs: hM4Di, $-1.39 \pm 0.14$ °C vs. mCherry, $-1.77 \pm 0.08$ °C, $p = 0.0370$; average ΔTc: hM4Di, $-1.37 \pm 0.11$ °C vs. mCherry, $-1.78 \pm 0.08$ °C, $p = 0.0139$) (Supplementary Fig. 7d, g–j). Because PBel contains highly heterogeneous populations of neurons, including neurons activated by warm or cool body temperature[42–44,52,53], the mild results in this experiment are likely due to activating different neurons of opposing functions.

To further confirm whether the direct PBel-PSTh pathway was critical for 2MT-evoked hypothermia, we specifically silenced the neurotransmission in the PSTh-projecting PBel neurons by inhibitory DREADDs. Briefly, the PSTh-projecting PBel neurons were labeled by bilateral injections of AAV2/retro-hSyn-Cre into the PSTh and AAV2/9-hSyn-DIO-hM4Di-mCherry or AAV2/9-hSyn-DIO-mCherry into the PBel, respectively (Fig. 7a, b). C21 treatment significantly blunted 2MT-induced tail temperature increase (average ΔTt: hM4Di, $-0.11 \pm 0.25$ °C vs. mCherry, $1.61 \pm 0.25$ °C, $p = 0.0007$) in the hM4Di-expressing mice relative to mCherry-expressing mice (Fig. 7c–e). Moreover, chemogenetic suppression of the PSTh-projecting PBel neurons also significantly attenuated 2MT-evoked hypothermia (average ΔTs: hM4Di, $-1.28 \pm 0.20$ °C vs. mCherry, $-2.26 \pm 0.26$ °C, $p = 0.0148$; average ΔTc: hM4Di, $-1.00 \pm 0.11$ °C vs. mCherry, $-1.83 \pm 0.31$ °C, $p = 0.0250$) (Fig. 7c, f–i). Taken together, these results strongly suggest that 2MT-activated PBel neurons project to PSTh to mediate 2MT-evoked hypothermia and tail vasodilation.

## Discussion

Emotions such as fear can trigger a rapid shift in thermoregulation in mammals as part of the "fight or flight" responses to dangerous situations. However, the neural mechanisms of fear-related thermoregulation remain largely unexplored. In this study, we showed that exposure to 2MT could induce rapid hypothermia accompanied by elevated tail temperature in wild-type but not *Trpa1*$^{-/-}$ mice. Based on our current studies (this paper and Kobayakawa's co-submitted paper)[30,54], we propose a working model for the neural mechanism underlying 2MT-evoked hypothermia (Fig. 8). Innate fear odor 2MT is sensed by the nociceptive receptor Trpa1, which is expressed on the surface of TG neurons and vagal ganglion (VG) neurons. Whereas VG neurons transmit the signal to NTS[55], TG neurons transmit the signal to PB and spinal trigeminal nucleus (Sp5) in the brain stem[56]. Because both TG and Sp5 neurons project to PB[41,57,58], we believe that a subpopulation of vGlut2$^+$ PBel neurons receive the 2MT-evoked innate fear signal directly from TG or indirectly via Sp5 neurons. Once activated, these vGlut2$^+$ PBel neurons transmit the signal to vGlut2$^+$ PSTh neurons, which then relay the signal to NTS neurons to trigger hypothermia and tail vasodilation.

In support of this model, optogenetic activation of the axon terminals or cell bodies of the NTS-projecting PSTh neurons or 2MT-activated, PSTh-projecting PBel neurons effectively induces hypothermia and tail vasodilation. Conversely, chemogenetic suppression of the vGlut2$^+$ excitatory neurons in the PBel or PSTh, or the PSTh-projecting PBel neurons attenuated 2MT-evoked hypothermia and tail vasodilation. Moreover, TeLC-mediated blockage of neurotransmission in the NTS-projecting PSTh neurons significantly blunted 2MT-evoked hypothermia and tail vasodilation. Thus, our results identify a PBel-PSTh-NTS neural pathway that mediates 2MT-evoked innate fear-associated hypothermia and tail vasodilation. Interestingly, Matsuo et al.[30] found that chemogenetic activation of the NTS-PB pathway could also induce hypothermia, suggesting the potential existence of a PB-PSTh-NTS-PB feedforward loop for 2MT-evoked hypothermia. Furthermore, they also reported that trigeminal, vagal, or olfactory ablation could partially suppress 2MT-evoked hypothermia[54]. While trigeminal neuron-specific Trpa1 knock-out attenuates, olfactory neuron-specific Trpa1 knockout did not affect the 2MT-induced hypothermia[54]. These results imply that the olfactory system might contribute to 2MT-induced hypothermia through Trpa1-independent mechanisms. Taken together, these results suggest that multiple neural pathways may work collaboratively to mediate 2MT-evoked hypothermia.

**PSTh is a nucleus for emotion-related thermoregulation.** The subthalamic nucleus (STN) is a well-known target for deep brain stimulation to treat motor deficiency in PD patients[46,47]. Interestingly, deep brain stimulation of STN can trigger either motor or non-motor effects in PD patients depending on the electrode position within STN[59,60]. Recent studies suggest that STN contains a subpopulation of neurons for processing

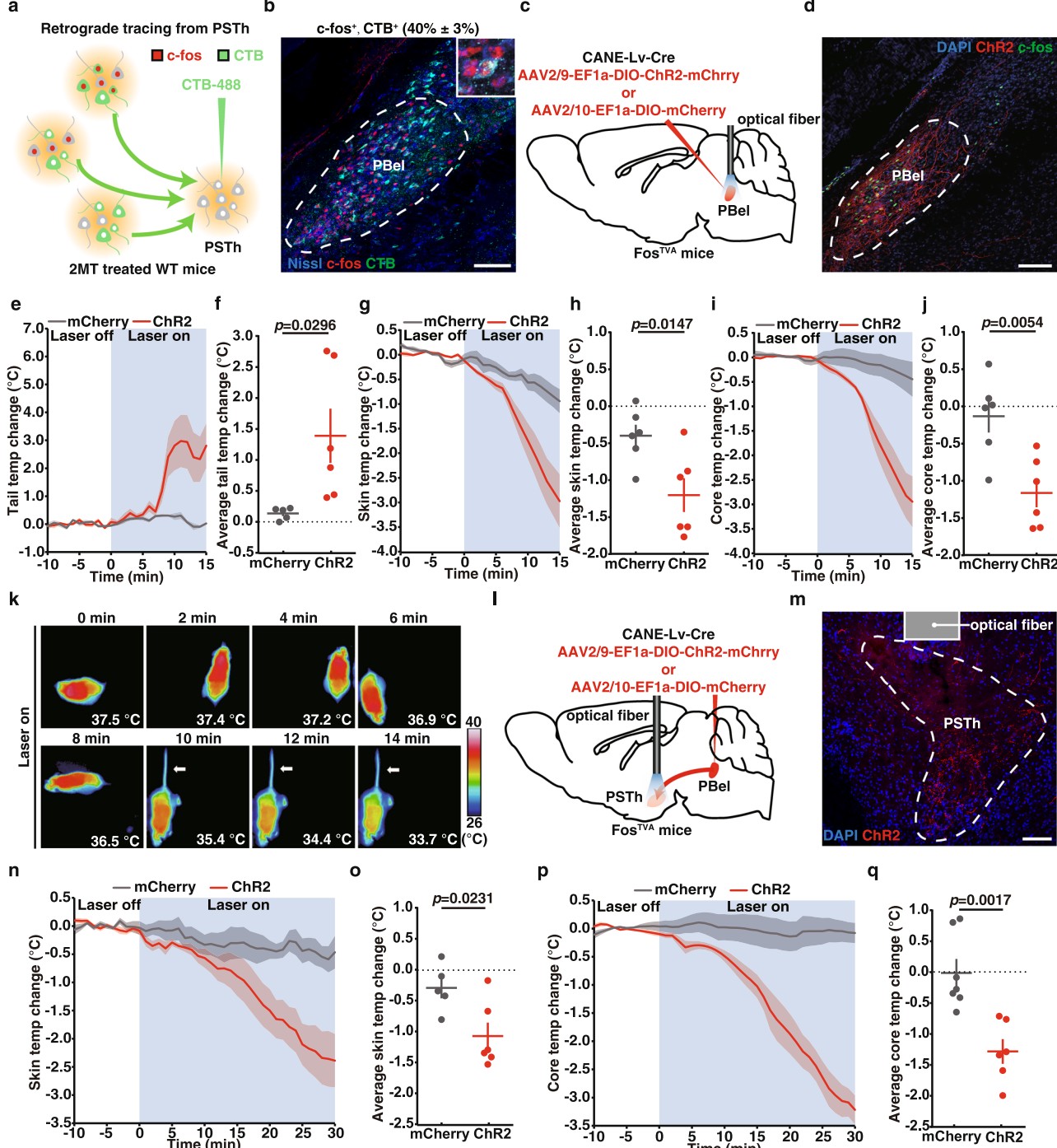

**Fig. 6 Activation of 2MT-activated PBel neurons and their axonal inputs to PSTh evokes hypothermia. a** Schematic of retrograde tracing from PSTh neurons by CTB. **b** Representative images showing double immunostaining of CTB and c-fos. The percentage of c-fos[+], CTB[+] neurons among CTB[+] neurons in PBel is shown in parenthesis ($n = 5$). **c** Schematic of selective opto-stimulation of the cell bodies of 2MT-activated PBel neurons labeled by CANE in Fos[TVA] mice. **d** Representative images showing c-fos expression in the ChR2-labeled PBel neurons following photoactivation ($n = 2$). Tail (**e**), skin (**g**), and core (**i**) temperature curves of mCherry-expressing ($n = 5$ or 7) and ChR2-expressing ($n = 6$) Fos[TVA] mice before and during blue light stimulation of 2MT-activated PBel neurons. Average tail (**f**), skin (**h**), and core (**j**) temperature changes of mCherry-expressing ($n = 5$ or 7) and ChR2-expressing ($n = 6$) Fos[TVA] mice during opto-stimulation of 2MT-activated PBel neurons. **k** Time-lapsed thermal images of ChR2-expressing Fos[TVA] mice during photoactivation of 2MT-activated PBel neurons. **l** Schematic of selective opto-stimulation of the axon terminals of 2MT-activated PBel neurons in PSTh in Fos[TVA] mice. **m** Representative images showing the ChR2-labeled axon terminals from 2MT-activated PBel neurons in PSTh ($n = 2$). Skin (**n**) and core (**p**) temperature curves of mCherry-expressing ($n = 5$ or 7) and ChR2-expressing ($n = 6$) Fos[TVA] mice before and during blue light stimulation of axon terminals in PSTh. Average skin (**o**) and core (**q**) temperature changes of mCherry-expressing ($n = 5$ or 7) and ChR2-expressing ($n = 6$) Fos[TVA] mice during blue light stimulation of axon terminals in PSTh. **e–j**, **n–q** Data are mean ± SEM; two-side Student's *t* test. Scale bars, 100 μm.

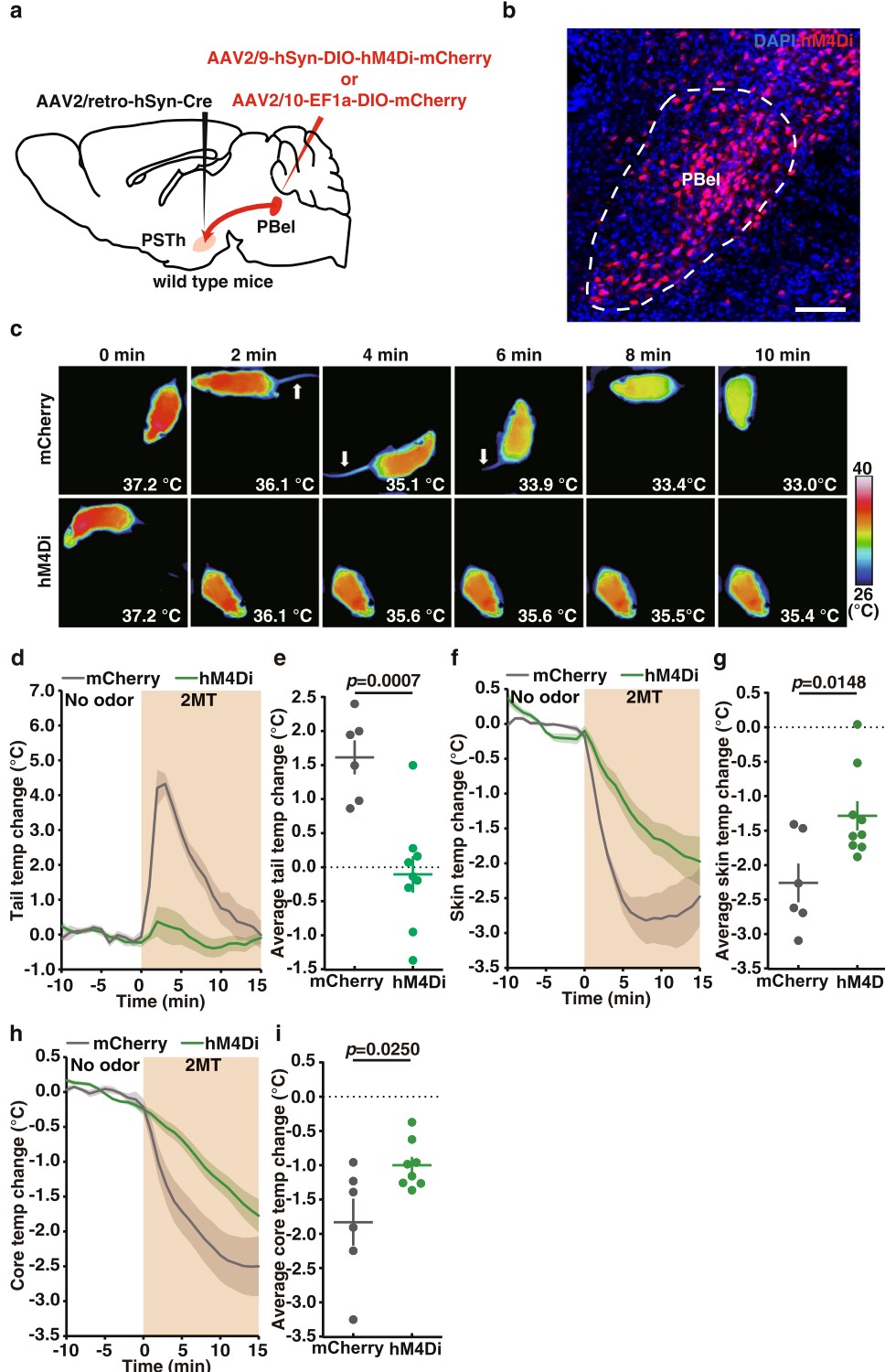

**Fig. 7 Inhibition of PSTh-projecting PBel neurons diminishes 2MT-evoked hypothermia and tail temperature increase. a** Schematic of chemogenetic inhibition PSTh-projecting PBel neurons. **b** Representative image of hMD4i-labeled PSTh-projecting PBel neurons (n = 2). **c** Time-lapsed thermal images of mCherry-expressing mice and hM4Di-expressing mice during 2MT treatment following administration of C21. Tail (**d**), skin (**f**), and core (**h**) temperature curves of mice with (hM4Di, n = 8 or 9) or without (mCherry, n = 6) inactivation of PSTh-projecting PBel neurons before and during 2MT treatment. Average tail (**e**), skin (**g**), and core (**i**) temperature changes of mice with (hM4Di, n = 8 or 9) or without (mCherry, n = 6) inactivation of PSTh-projecting PBel neurons during 2MT treatment. **d–i** Data are mean ± SEM; two-side Student's t test. Scale bar, 100 μm.

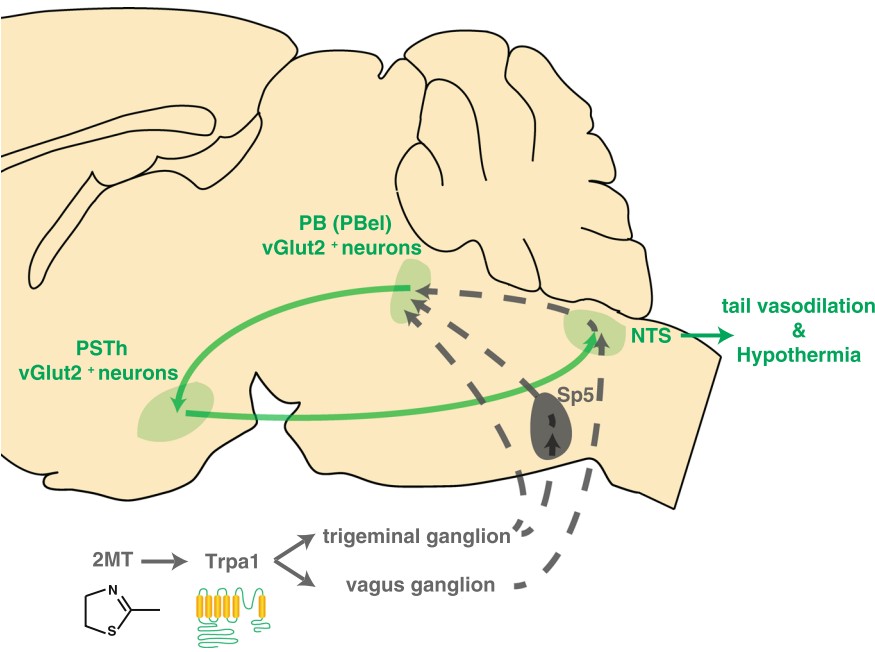

**Fig. 8 Neural pathways for 2MT-evoked hypothermia.** 2MT is sensed by Trpa1 in trigeminal ganglion (TG) neurons and vagal ganglion (VG) neurons. A subpopulation of vGlut2[+] PBel neurons receive the 2MT signal directly from TG neurons or indirectly via Sp5 neurons. These vGlut2[+] PBel neurons transmit the 2MT signal to vGlut2[+] PSTh neurons, which relay the signal to NTS neurons to trigger hypothermia and tail vasodilation.

emotion-related information[61–63]. Moreover, the single-neuron recording has identified these STN neurons that respond to emotional stimuli, such as the presentation of affective pictures[64]. Here, we observed that exposure to innate fear odor 2MT dramatically induced c-fos expression in the posterior part of STN, called PSTh (Fig. 1i, j), although it is unclear whether these 2MT-activated PSTh neurons are the same STN neurons that respond to emotional stimuli. Because PSTh neurons receive direct axonal projections from 2MT-activated PBel neurons (Fig. 6b, m), it is plausible that these PSTh neurons are involved in the processing of 2MT-evoked innate fear signals. Consistent with our hypothesis, it has recently been shown that the CGRP[+] neurons in PBel, which serves as a general alarm system, directly project to PSTh[48]. Furthermore, we found that 2MT-activated PSTh neurons directly project to NTS (Fig. 2b), a critical node for autonomic regulation and thermoregulation, further supporting our hypothesis that PSTh contains a neural population responsible for emotion-related thermoregulation. Thus, our studies identify PSTh as a thermoregulatory hub that connects PBel to NTS to mediate innate fear-associated hypothermia. Future studies are warranted to investigate whether PSTh is also involved in other emotion-related and patho/physiological thermoregulatory processes.

**How does PSTh-NTS regulate tail vasodilation?** The two major autonomic responses to reduce body temperature are the suppression of BAT-dependent thermogenesis and facilitation of heat dissipation from glabrous skin, such as from the tail in rodents[3]. Matsuo et al.[30] recently discovered that 2MT-evoked hypothermia was independent of suppression of BAT thermogenesis. In this study, we found that 2MT exposure induced a sharp (up to 6 °C) increase in the tail temperature accompanying the rapid hypothermia in wild-type mice (Fig. 1b–h). Because it is well-known that vasodilation induces heat dissipation from the skin[1,3], we expect that heat dissipation from the mouse tail contributes critically to 2MT-evoked hypothermia.

The vasomotor is controlled by the spinal sympathetic preganglionic neurons through sympathetic ganglion neurons in the brain stem. The rostral raphe pallidus nucleus (rRpa) is the core nucleus regulating the activity of sympathetic preganglionic neurons for controlling cutaneous vasculature, and inhibition of rRpa results in vasodilation[65]. Alternatively, the RVLM is proposed as part of the presynaptic nuclei of spinal sympathetic preganglionic neurons for regulating cutaneous vasomotor control[3,49]. Activation of RVLM elicits vasoconstriction and increases blood pressure[49,66], whereas suppression of RVLM activity induces vasodilation[49]. Moreover, excitation of inhibitory neurons in caudal ventrolateral medulla induces vasodilation through suppression of the neural activity in RVLM[49]. Future studies are warranted to determine precisely which of these vasomotor controlling nuclei may function downstream of the PBel-PSTh-NTS pathway to mediate 2MT-evoked tail vasodilation.

**Why does 2MT, but not cinnamaldehyde, induce hypothermia?** Although 2MT and cinnamaldehyde are both Trpa1 agonists, 2MT but not cinnamaldehyde could induce hypothermia in wild-type mice, suggesting that the activation of Trpa1 is necessary but not sufficient to induce hypothermia. In a companion study[54], Kobayakawa and colleagues performed a series of nice experiments to clarify the discrepancy of why 2MT, but not cinnamaldehyde, induces hypothermia in mice. Remarkably, in vivo calcium imaging showed that 2MT and cinnamaldehyde could evoke different kinetics of calcium influx in TG neurons, suggesting a distinct mode of Trpa1 activation. RNA-seq analysis revealed distinct gene expression profiles in the TG of wild-type mice following 2MT or cinnamaldehyde exposure. Moreover, 2MT, but not cinnamaldehyde, could induce high c-fos expression in the Sp5 neurons, which receive synaptic input from Trpa1[+] TG neurons. Importantly, chemogenetic activation of the Sp5-projecting Trpa1[+] TG neurons is sufficient to induce moderate hypothermia. These results, together with the observation that the inactivation of Trpa1 abolishes 2MT-evoked hypothermia, strongly suggest that Trpa1 is the chemosensor for 2MT-evoked hypothermic response. It is worth noting that the activation of Trpa1 by different agonists is more complex than a

simple binary response, and only specific Trpa1 agonists like 2MT could evoke rapid hypothermia in mice. Future studies are needed to further investigate the detailed mechanisms by which different agonists differentially activate Trpa1 in the TG and/or VG neurons to induce distinct cellular and/or physiological responses.

**Biological significance of innate fear-associated hypothermia.** Different types of stress or fear stimuli trigger a combination of defensive behaviors and physiological responses, including thermoregulation, to promote animal survival in perceived dangerous conditions. In general, stress/fear-induced hyperthermia is a far more common phenomenon than stress/fear-induced hypothermia. For instance, short duration of physical stress, such as social defeat, tail pinch, and restrain stress, induces hyperthermia[7]. Both learned fear and mild innate fear stimuli, such as ferret or fox odorants, induce hyperthermia in rodents[20–22]. On the other hand, potent stress or fear stimuli that threaten the survival of animals, such as long-lasting immobile stress and hypoxia, induces hypothermia[7,23–25]. Moreover, potent innate fear odor 2MT can trigger acute hypothermia accompanying robust defensive behaviors, such as freezing[28–30]. Thus, it is plausible that extreme stress or fear stimuli may induce hypothermia to promote animal survival, although the biological significance of stress/fear-evoked hypothermia is unclear. It has recently been shown that 2MT exposure induces potent bioprotective effects against hypoxia and ischemia/reperfusion injuries of the skin and brain by inducing hypothermia, anaerobic metabolism, and anti-inflammatory effects[30]. Thus, innate fear/stress-associated hypothermia may have evolved as a physiological defensive strategy against potent threats for animal survival.

## Methods
**Animals.** Animal protocols used in this study were approved by the International Institute for Integrative Sleep Medicine at the University of Tsukuba, Japan and by the National Institute of Biological Sciences, Beijing, China. Mice were housed in groups of 4–5 and maintained on a 12-h light–dark schedule (lights on at 9:00) with ad libitum access to food and water. Ambient temperature was 23.5 °C and humidity was controlled at 40–60%. C57BL/6J mice were obtained from CLEA, Japan. *Trpa1*$^{-/-}$ (stock 006401), Fos$^{TVA}$ (stock 027831), and *vGlut2-IRES-Cre* (stock 016963) mice were purchased from the Jackson Laboratory (Bar Harbor, ME), United States. We used ≥8-week-old male mice for all experiments.

**Body temperature recording.** The skin and core body temperatures were recorded by IR digital thermographic camera (H2640, NIPPON AVIONICS) or by telemetry transmitter (TA11TA-F10, DSI), respectively. To measure the skin temperature, the back hair was shaved under anesthesia with isoflurane 1 day before experiments. Skin temperature was continuously recorded at 1 frame/s, and the highest temperature in one frame was used as the skin temperature of the animal for analysis according to the user manual (InfRec Analyzer; NIPPON AVIONICS). We manually analyzed the tail temperature every 60 frames. The tail temperature is measured at the point 1 cm away from the base of the tail. To measure the core body temperature, the telemetry transmitter was implanted in the peritoneal cavity under anesthesia with isoflurane more than 7 days before experiments. The core body temperature was continuously measured upon 2MT (Tokyo Chemical Industry) exposure or optogenetic stimulation. Test mice were singly housed 1 day before 2MT experiment. An individual mouse was habituated for 30 min in the temperature recording cage (17.5 × 10.5 × 15 cm) placed in a fume hood. After 10 min with no odor exposure, a filter paper soaked with 20 μl 2MT was placed at the center of the cage for 15 min. All temperature recording experiments were performed under ambient temperature at 23.5 °C.

**2MT or cinnamaldehyde exposure.** The mice were singly housed 1 day before 2MT (Tokyo Chemical Industry) or cinnamaldehyde (Nacalai Tesque) exposure experiment. An individual mouse was habituated in a temperature recording cage (17.5 × 10.5 × 15 cm) for 30 min. After 10 min with no odor exposure, 20 μl of 2MT (2.1 × 10$^{-4}$ mole) or cinnamaldehyde was dropped on the filter paper (~4.0 cm$^2$) and placed at the center of the cage for 15 min. For c-fos immunostaining, a 2MT (20 μl) soaked filter paper was placed in the mouse's home cage for 120 min before collecting the brain samples.

**Viruses.** The following viruses were used for experiments: AAV2/9-hSyn-DIO-hM4Di-mCherry (Addgene, 44362), AAV2/9-EF1a-DIO-ChR2-mCherry[67],

AAV2/10-EF1a-DIO-mCherry (Addgene, 50462), AAV2/retro-hSyn-ChR2-EYFP (Addgene, 26973), AAV2/Retro-hSyn-EGFP (Addgene, 50465), AAV2/9-hSyn-FLEX-TeLC-P2A-GFP[68], AAV2/9-hSyn-FLEX-EGFP (Addgene, 50457), AAV2/retro-hSyn-Cre (Addgene, 122518), CANE-LV-Cre[51]. Both AAV and lentivirus were produced as previously described[51,67].

**Stereotaxic injections and implantation of optical fiber.** Mice were anesthetized with 1.5–2% isoflurane and placed in a stereotaxic frame (David kopf Instruments), and small craniotomies were made over the target brain regions. Stereotaxic coordinates of virus/CTB injection were as follows: PSTh, AP −2.00 mm, ML ± 1.20 mm, DV −5.10 mm; PBel, AP −4.80 mm, ML ± 1.85 mm, DV −4.00 mm; NTS, AP −7.80 mm, ML ± 0.30 mm, DV −4.50 mm. To reach the final injection site of PBel and NTS, a small craniotomy was made at the following coordinates: PBel, AP −4.10 mm, ML ± 1.85 mm, DV −4.00 mm; NTS, AP −7.00 mm, ML ± 0.30 mm, DV −5.70 mm. In addition, the virus/CTB injection pipette was back toward bregma at a 10 degree angle relative to vertical to inject into PBel and NTS. The virus or CTB was injected at a slow flow rate (100 nl/min) by using a pulled thin glass pipette to avoid potential damage in the injection site. The pipette was withdrawn at least 10 min after viral injection. For implantation of optical fibers (custom made: diameter, 200 μm; NA, 0.22), implants were lowered above the target sites and secured to the skull with Resin Cement (3 M RelyX Unicem 2). The coordinates of optical fiber implantation were as follows: PSTh, AP −2.00 mm, ML ± 1.20 mm, DV −4.50 mm; PBel, AP −4.10 mm, ML ± 1.85 mm, DV −3.80 mm; RNTS, AP −6.20 mm, ML ± 0.3 mm, DV −4.40 mm; CNTS, AP −7.00 mm, ML ± 0.4 mm, DV −4.50 mm. The optical fiber was back toward bregma at a 10 degree angle relative to vertical to implant into PBel and NTS.

**Capturing 2MT-activated neurons in PBel using the CANE method.** To stably express ChR2 in 2MT-activated PBel, singly housed Fos$^{TVA}$ mouse was exposed 2MT (20 μl with filter paper) for 15 min. Around 45 min later, the Fos$^{TVA}$ mouse was lightly anesthetized with isoflurane and fixed in stereotaxic flame. By following the previously described virus injection method, the mixture of CANE-Lv-Cre and AAV2/9-EF1a-DIO-ChR2-mCherry or AAV2/10-EF1a-DIO-mCherry was injected into PBel.

**Optogenetic and chemogenetic experiments.** To activate ChR2-labeled neurons, blue light was delivered by a laser (473 nm; Shanghai Laser & Optics Century). Before optogenetic stimulation, the mice were lightly anesthetized with isoflurane and a ferrule patch cord (Doric Lenses) was connected to ferrule optic fiber. The laser pulses were controlled through Doric Neuroscience Studio. The mice were photostimulated at 20 mW, 20 Hz, and 25 ms pulse duration for 15-min stimulation pattern on neurons and for 30-min stimulation pattern on axon terminals. For chemogenetic experiment, hM4Di or mCherry-expressing mice were injected i.p. with C21 (1 mg/kg; HB6124, Hello Bio) 30 min before 2MT exposure.

**Immunohistochemistry, image acquisition, and quantification.** Mice were deeply anesthetized with isoflurane and transcardially perfused with ice-cold 10% sucrose in Milli-Q water, followed by ice-cold and 4% paraformaldehyde (PFA) in 0.1 M phosphate buffer saline, pH 7.4 (PBS). Dissected brains were postfixed for overnight in 4% PFA and transferred to 30% sucrose in PBS at 4 °C. Then, brains were frozen in Tissue-Tek O.C.T (Sakura) and stored at −80 °C. Coronal brain slices (80 μm) were made using a cryostat (Leica Biosystems). After washing with PBS three times, brain slices were treated with permeabilization solution (1% Triton in PBS) at room temperature for 3 h and followed by incubation with the blocking buffer (10% Blocking One (Nacalai Tesque) with 0.3% Triton X-100 in PBS) at room temperature for 1 h. Brain slices were incubated with the primary (first) antibody in blocking buffer at 4 °C overnight. After washing with PBS three times, brain slices were incubated with secondary (second) antibody in blocking buffer at 4 °C overnight. The slices were stained with NeuroTrace fluorescent Nissl stain (Invitrogen, N-21279) or 4′, 6-diamidino-2-phenylinodole (DAPI) (Dojindo, D523), washed, mounted, and coverslipped. Immunostained brain slices were imaged using a Zeiss LSM800 confocal microscope with a ×10 or ×20 objective lens by Zeiss software Zen. Antibodies information: first antibodies: 1:5000 Rabbit polyclonal anti-c-fos (Sigma-Aldrich, ABE457), 1:4000 Goat polyclonal anti-mCherry (SCIGEN, AB0040–200), 1:4000 Rat monoclonal anti-GFP (Nacalai tesque, 04404–84); all second antibodies were obtained from Jackson ImmunoResearch: 1:1000 Donkey anti-rabbit Alexa Fluor 488 (711-545-152), 1:1000 Donkey anti-rabbit Cy3 (711-165-152), 1:1000 Donkey anti-rat Alexa Fluor 488(712-545-153), 1:1000 Donkey anti-goat Cy3 (705-165-147).

**Cell counting and quantification.** Images were analyzed by Image J or Adobe Photoshop counting tool. For the c-fos mapping experiments, the cell density of c-fos positive neurons in each region was calculated. In each brain region, we randomly selected three to five brain slices and manually outlined the target nucleus. The area of the target nucleus was calculated automatically by Image J. Then, we counted the number of c-fos positive neurons inside the outline and calculated the cell density of c-fos positive neurons by dividing the area in each section. The final cell density of c-fos positive neurons in each region was the average cell density among sections containing the target region in each mouse.

Quantification of the two-color fluorescence in situ hybridization results (the cell number and the ratio of c-fos$^+$, vGlut2$^+$ or c-fos$^+$, vGAT$^+$ double-positive neurons) and the number of CTB-based retrograde labeling results (the cell number and the ratio of c-fos$^+$, CTB$^+$ double-positive neurons) were performed as same as c-fos mapping experiments described above. The final percentage of each population was the average percentage among sections containing the target region in each mouse. To quantify the ratio of TeLC-expressing neurons in PSTh, the number of TeLC+ neurons and the total number of DAPI signals in PSTh were similarly counted as described above. The infection rate of each mouse indicates the percentage of TeLC$^+$ neurons in PSTh.

**Two-color in situ hybridization**. The cDNA fragments of mouse c-fos, vGlut2, and vGAT genes were amplified by PCR with antisense primers containing T7 promoter sequence. In vitro transcription was performed with PCR-amplified templates using T7 RNA polymerase (Roche) to synthesize antisense RNA probes (Supplementary Table). Two-color in situ hybridization was performed based on a basic method[69]. Briefly, mice were exposed 2MT for 30 min, deeply anesthetized with isoflurane, and transcardially perfused with ice-cold 10% sucrose in Milli-Q water followed by ice-cold 4% PFA. The collected brain samples were postfixed with 4% PFA at 4 °C overnight, followed by displacement with 30% sucrose in PBS containing 0.1% diethylpyrocarbonate at 4 °C overnight. Coronal brain sections (40 μm) were made using a cryostat (Leica Biosystems). Brain sections were treated with proteinase K (Roche) followed by acetylation, and then incubated with hybridization buffer containing antisense RNA probes at 60 °C for 16 h. After stringent washing, brain sections were incubated with horseradish peroxidase (HRP)-conjugated anti-FITC antibody (PerkinElmer; 1:1000) or HRP-conjugated anti-Dig antibody (Roche; 1:1000) at 4 °C overnight. To sequentially use HRP-conjugated antibody, 2% sodium azide solution was treated for the inactivation of HRP. TSA system (TSA-FITC and TSA-Biotin; PerkinElmer) and Streptavidin Alexa Fluor 568 conjugate (ThermoFisher) were used to visualize the mRNA signals.

**Statistics**. Microsoft Excel and GraphPad Prism 6 were used to analysis statistics and make graphs. For statistical analyses of the experimental data, two-sided Student's t test, two-way ANOVA analysis, and linear regression analysis were used. The n used for these analyses represents the number of mice. Detailed information on statistical analyses is provided in the figure legends.

**Reporting summary**. Further information on research design is available in the Nature Research Reporting Summary linked to this article.

## Data availability
Source data are provided with this paper. The relevant data are available from the corresponding authors upon reasonable request.

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

## Acknowledgements

We thank Dr. Yoan Cherasse for sharing reagents, Drs. Takeshi Sakurai and Michael Lazarus for comments on our paper. This work was supported by the Beijing Municipal Science & Technology Commission (Z181100001318004 to Q.L.) and the National Key Research and Development Program of China, and by the JSPS KAKENHI grants (15K14874 and 17H06048 to Q.L.; 18K06515, 18H04684, and 19H05006 to K.S.; 17K08133 and 20K06738 to L.C.; 18H04806 to K.K.; 18H02546 and 20H04849 to R.K.), Canon Foundation (to K.K.), the Takeda Science Foundation (to R.K. and K.K.), and the World Premier International Research Center Initiative (WPI) from MEXT, Japan.

## Author contributions

Q.L., K.S., and C.L. designed the experiments with help from K.K., R.K., and P.C. C.L., C.-Y.L., and G.A. conducted comparative c-fos mapping, whereas C.L., C.-Y.L., L.C., and K.S. performed thermo-imaging and analysis. C.L. and C.-Y.L. were responsible for all optogenetic/chemogenetic manipulation and CTB tracing experiments. Y.T. and K.S. conducted immunohistochemistry and in situ hybridization experiments. Y.T. took care of mouse husbandry and K.S. produced AAV and Lentivirus. C.L. prepared the all figures and Q.L., K.S., C.L. wrote the paper.

## Competing interests

The authors declare no competing interests.
