## [Peer Review File · Nature Communications]

Reviewer #1 (Remarks to the Author):

The manuscript by Liu et al uncovered a novel lateral parabrachial (PBel) to posterior subthalamic nucleus (PSTh) to nucleus of solitary tract (NTS) pathway that mediate the predator-odor (2MT) induced hypothermia and tail vasodilation in mice.

Overall this study is very interesting, and conclusions are largely supported by both gain- and loss-of-function experiments. However, there are a few points that need to be addressed before the manuscript is acceptable for publication:

1. The authors used AAVretro-Cre in NTS, and Cre-dependent virus in PSTh to express Chr2/EGFP/TeLC in PSTh-to-NTS neurons. However, PSTh neurons likely project to many other targets (in addition to NTS).

It is important for the authors to show what are the other targets from PSTh neurons labeled using this method as these other targets maybe important in regulating temperature.

The authors never directly manipulated the axonal terminal activity of PSTh neurons in NTS. It would greatly enhance the conclusions of this study if the authors could place optic fiber directly in NTS and stimulate axon terminals of PSTh-to-NTS neurons. This reviewer understands that NTS is a rather large area and it may be inefficient to use optic fiber to excite enough axon fibers. However, people in the field working on projections to NTS have been able to find the axonal bundle near NTS i.e. before they enter into NTS and arborize), and stimulate the bundle rather than terminals and get strong effects. Therefore, I recommend the authors to at least try this experiment.

2. In Figure 5, while it was elegant to use Fos-based method (CANE) to selectively target 2MT activated PBel neurons, Chr2-stimulating axonal terminals of PBel in PSTh was significantly slower and less efficient at inducing skin/tail temperature changes compared to stimulating the cell bodies. The authors attributed this to low efficiency labeling using CANE or inability to activate sufficient number of axons.

However, an alternative explanation is that PSTh is NOT the only target of PBel that are involved in 2MT-induced temperature change as it is known that PBel neurons project to multiple forebrain areas. It could be useful to show other axonal projection areas of the CANE-labeled PBel neurons, and then state the possibility that these other areas could also be involved.

3. After using the elegant CANE method to selectively label 2MT-activated PBel, the authors switched to the non-specific vGlut2-Cre to inhibit PB neurons using hM4Di in Figure 6. This is a questionable experiment. PB contains highly heterogeneous populations of neurons, including neurons activated by warm or cool body temperature (e.g. Geerling et al., *Am. J. Physiol. Regul. Integr. Comp. Physiol.*, 2016). Not surprisingly, the results shown in Figure 6 is mild at best, likely due to activating neurons of opposing functions.

If CANE is inefficient for performing inhibition experiments, the authors should at least use AAVretro-Flex-hMDi injected into PSTh and Cre virus injected into PBel, to express hM4Di-DREADDs in PBel neurons projecting to PSTh, and perform inhibition experiment.

4. The authors suggest 2MT is sensed by TG or VG, and TG project to Sp5 and Sp5 project to PB. However, a recent paper (Rodriguez et al, *Nature Neurosci.* 2017) showed some TG neurons send monosynaptic projection directly to PB. The authors should revise their model to include this direct connection.

Reviewer #2 (Remarks to the Author):

The authors have performed a series of retrograde tracing and optogenetic and dREADT experiments in TRPV1 KO and WT mice to discover the neural circuitry underlying the cutaneous vasodilatory and hypothermic response observed when mice are exposed to the scent of 2MT. The overall relevance to “fear” responses in animals other than mice or to stimuli other than 2MT, as well as several components of the proposed neural circuit for this hypothermia remain to be determined. The main novel insight provided by this study is the discovery that the PSTh contains neurons that can elicit a significant effect on T_{core} through inhibition of cutaneous vasoconstriction.

Relevance: The authors attempt to couch the relevance of their study of this 2MT-evoked hypothermia/vasodilation in the framework of “innate fear”, as opposed to “stress”, which is accompanied by hyperthermia. However, they provide no teleological explanation or testable hypotheses for this thermoregulatory response to the odor of 2MT, nor are we provided with any examples in other species of a predator odor-driven hypothermia, or in general of a fight, flight or freeze situation that induces hypothermia. Overall, this anthropomorphic assignment of “fear” to this mouse vs predator encounter may not be valid, since no data are provided to show that a similar cutaneous vasodilation and hypothermia are evoked when a mouse is exposed to a predator whose odor is not an agonist of TRPA1. Thus, it might be more accurate to suggest that they are studying TRPA1-evoked hypothermia. To maintain this “fear” framework for their studies, the authors should demonstrate that other TRPA1 agonists, which are not predator odors, do not evoke a similar cutaneous vasodilation and hypothermia. It may simply be that this particular mouse predator odor happens to be a TRPA1 agonist.

Nonetheless, these elegant experiments do reveal novel connections between the PBel, the PSTh and the NTS that contributes to TRPA1-evoked hypothermia and tail vasodilation in mice (the authors make a similar, albeit over-interpreted, conclusion in their Abstract). This is not a “circuit” since neither the sensory component (sensing the TRPA1 agonist) nor the primary efferent component (sympathetic premotor neurons for cutaneous vasoconstriction in the tail) are among these 3 synaptic sites, and the afferent and efferent connections to these 3 sites were not studied.

Approach: The powerful molecular/genetic techniques employed in this study significantly strengthen the authors’ conclusions regarding the connections among these 3 nuclei, as well as their functional connection to the 2MT stimulus. As the authors point out, the inability of the stimulation and inhibition experiments to completely mimic the magnitude of the original, 2MT stimulus or to completely block it could be the result of incomplete transduction or limited laser light exposure of relevant neurons – both of which are somewhat beyond the control of the investigators.

Unfortunately, these ‘incomplete’ responses could also suggest that these 3 brain regions are not the only ones involved in the 2MT-evoked hypothermia. In this regard, the authors performed several anatomical (tracing, c-fos) experiments, but only report “selected” results. In the middle of page 6, for instance, the authors indicate that the PBel, PSTh and NTS had more c-fos⁺ neurons in Trpa1^{-/-} mice, but they do not provide a table of the c-fos⁺ neuron counts in the various brain nuclei examined in WT and KO mice, so we do not know how extensive their survey of c-fos⁺ neurons was, or whether other nuclei also had more c-fos⁺ neurons in the KO mice, suggesting a possible role in the hypothermic/vasodilatory response. In addition, the authors should mention the caveat that neurons that are inhibited during the 2MT response (these include at least the sympathetic premotor neurons controlling tail vasoconstriction) will not be c-fos⁺. Similarly, it is hard to interpret the data provided at the bottom of page 7 – what percent of c-fos⁺ neurons were also CTB⁺? This would indicate whether activated PSTh neurons might also project to other targets. Again, on the

top of page 12, the other projection sites of 2MT-activated PBel neurons should be reported. The data with dreadd inhibition of vGlut2+ neurons in PBel (page 13) are not interpretable, since this population of neurons is quite functionally diverse, with different projection targets, at least one of which (to the POA) plays a significant role in normal thermoregulatory responses.

Altogether, the authors' data only allow the conclusion that neurons connected among the 3 sites "contribute", or "are involved in" the 2MT-evoked response in mice. Considering the already demonstrated significant roles of the PBel and the NTS in thermoregulatory and BAT thermogenic control in rats, it seems difficult to believe that the neurons identified here in mice are not also involved in some aspect of normal thermoregulation or metabolic homeostasis, but the circuits for normal thermoregulatory responses have not been delineated in mice. It would have been very informative to confirm, for instance, that the PBel neurons identified with 2MT exposure are a separate population from those responding to cold exposure and projecting to the POA.

A shortcoming of this study is the absence of data on the activity of the mice during various manipulations. This is directly relevant since mice are highly dependent on activity thermogenesis to maintain their T_{core} in a subthermoneutral environment (testing in a thermoneutral experimental condition could have been informative in this study). If 'freezing' (i.e., no somatic movement) is maintained for several minutes, one would expect T_{core} to fall, independent of tail vasodilation or inhibition of BAT thermogenesis. Thus, if the 3 brain regions identified here are involved in inhibiting tail vasoconstriction, but not in inhibiting somatic movement, one might expect that activation of any one of these sites would completely recapitulate the cutaneous vasodilatory response, but not the hypothermia evoked by 2MT exposure. This was exactly the case with the TeLC silencing of PSTh neurons (page 9).

Interpretation: The authors should discuss other functions of TRPA1-expressing afferents, since the life-long, global TRPA1 KO could affect a variety of other tonic and phasic reflexes involved in the homeostatic regulation of other, non-thermoregulatory systems (e.g., cardiovascular, respiratory) that might interfere or abrogate the responses to 2MT.

The author should also indicate the proposed route for the 2MT molecules to gain access to the TRPA1 on the sensory terminals of trigeminal and vagal neurons. How have they ruled out the potential for these afferents to be providing a tonic input to the normal thermoregulatory pathways that is unrelated to sensing of 2MT, but is required for the elaboration of the 2MT cutaneous vasodilation?

Curiously, the authors seem unaware of the significant literature on the CNS pathways specifically regulating cutaneous vasoconstriction! Due to its specific role in thermoregulation, blood flow in this vascular bed is controlled quite differently from that in other vascular beds. Rather than reference #46, the authors should study the work of Robin McAllen and of William Blessing, including their excellent review in *Comprehensive Physiology*. Sympathetic premotor neurons for cutaneous vasoconstriction are primarily contained in the medullary raphe pallidus nucleus, and it is the inhibition of these neurons that is likely involved in the cutaneous vasodilatory response to 2MT. Thus, the discussion of this issue on pages 16-17 should be completely rewritten.

Reviewer #3 (Remarks to the Author):

This is an interesting paper that demonstrates, and explores the regulation of, 2-methyl-2-thiazoline (2MT) induced hypothermia and elevated tail temperature in mice. A variety of methods are used including studies in mice lacking *Trpa1*, the chemosensor for 2MT, *c-Fos* to trace potential neural

regulatory pathways, optogenetic activation and tetanus toxin light chain (TeLC)-mediated inactivation of neurons with identified areas. A pathway including the posterior subthalamic nucleus (PSTh), external lateral parabrachial subnucleus (PBel), and nucleus of the solitary tract (NTS) are implicated in regulating odor fear mediated hypothermia, and a proposed circuit is provided. The methods and analyses appear appropriate. A few suggestions are provided below.

Major

As written, the manuscript seems to imply that hypothermia is the more typical stress/fear response; however, hyperthermia is induced by multiple stressors, and fear, stress and emotion can produce the well-known stress-induced hyperthermia response. Thus, a discussion that compares different thermoregulatory responses and speculation as to how different fearful stimuli produce different thermal responses would be useful and very interesting. At the very least, a greater acknowledgement of the different responses should be provided. An example is that inescapable stress, mentioned in the Introduction and Discussion, needs to be specified because some types of inescapable stress, as well as escapable stress, produce hyperthermia. Restraint, which was cited as producing hypothermia, has also been reported to produce hyperthermia.

Minor

Seven days is a brief recovery period for telemetry implants. Is there a possible effect of recovery from surgery in the results? Also, how long were the mice allowed to recover from anesthesia prior to optogenetic stimulation?

How was Adobe Photoshop used to analyze images?

Preparation of the filter paper with TMT should be provided for both behavioral and c-fos studies.

Overall, the paper reads well; however, it need significant editing with respect to grammar, particularly verb tense and the use of singular and plural nouns.

We thank all three reviewers for their positive comments and constructive criticisms. In the last four months, we have worked very hard on multiple sets of new experiments to address all of the reviewers' concerns. In the revision, we have added six major pieces of new experimental data that, together with previous results, firmly established that **“Posterior subthalamic nucleus (PSTh) mediates innate fear-associated hypothermia”**.

Added six new experimental data in the revision:

1. Exposure to Cinnamonaldehyde, a well-known TRPA1 agonist, could not induce hypothermia;
2. Comprehensive c-fos mapping of different brain regions in WT and *Trpa1* KO mice, including known fear/stress, thermoregulation, and vasodilation/vasoconstriction centers;
3. Optogenetic stimulation of the axon terminals of PSTh neurons in rostral NTS, but not in caudal NTS, causes robust hypothermia and tail temperature increase;
4. CANE-labeled PBel neurons do not project to POA;
5. Optogenetic stimulation of the axon terminals of CANE-labeled PBel neurons in CeA causes a modest reduction in skin and body temperature;
6. Chemogenetic inhibition of PSTh-projecting PBel neurons significantly suppressed 2MT-evoked hypothermia and tail vasodilation;

A point-by-point response to reviewers' comments is listed below. The revised parts of the manuscript are highlighted in blue.

Reviewer #1:

The manuscript by Liu et al uncovered a novel lateral parabrachial (PBel) to posterior subthalamic nucleus (PSTh) to nucleus of solitary tract (NTS) pathway that mediate the predator-odor (2MT) induced hypothermia and tail vasodilation in mice. Overall this study is very interesting, and conclusions are largely supported by both gain- and loss-of-function experiments. However, there are a few points that need to be addressed before the manuscript is acceptable for publication:

RE: We sincerely thank Reviewer 1 for recognizing the importance of our work and our approach.

1. The authors used AAVretro-Cre in NTS, and Cre-dependent virus in PSTh to express ChR2/EGFP/TeLC in PSTh-to-NTS neurons. However, PSTh neurons likely project to many other targets (in addition to NTS). It is important for the authors to show what are the other targets from PSTh neurons labeled using this method as these other targets maybe important in regulating temperature.

RE: We thank Reviewer1 for pointing this out. We performed two experiments: 1) Injection of AAVretro-hSyn-ChR2/EGFP in NTS; 2) Injection of AAVretro-Cre in NTS and Cre-dependent TeLC/EGFP virus in PSTh, to express ChR2/EGFP/TeLC in the NTS-projecting PSTh neurons. In both experiments, however, we did not find other projection targets from the NTS-projecting PSTh neurons.

The authors never directly manipulated the axonal terminal activity of PSTh neurons in NTS. It would greatly enhance the conclusions of this study if the authors could place optic fiber directly in NTS and stimulate axon terminals of PSTh-to-NTS neurons. This reviewer understands that NTS is a rather large area and it may be inefficient to use optic fiber to excite enough axon fibers. However, people in the field working on projections to NTS have been able to find the axonal bundle near NTS i.e. before they enter into NTS and arborize), and stimulate the bundle rather than terminals and get strong effects. Therefore, I recommend the authors to at least try this experiment.

RE: We thank Reviewer1 for this great suggestion. We performed exactly this experiment to confirm whether the PSTh-NTS pathway was involved in 2MT-induced hypothermia. Briefly, we specifically labeled PSTh neurons by injecting AAV expressing Cre-dependent ChR2 into the PSTh, and implant optic fibers onto the NTS of *vGlut2-Cre* mice. Because NTS is a rather large area, we implanted optic fibers on either the rostral or caudal side of NTS to distinguish which part of NTS is responsible for 2MT-induced hypothermia. Interestingly, we found that optogenetic stimulation of the axon terminals of PSTh neurons in rostral NTS (RNTS), but not caudal NTS (CNTS), had a more profound effect in tail temperature increase and body temperature reduction. These new results further confirm that the PSTh-NTS pathway plays a crucial role in 2MT-evoked hypothermia (new Fig 5).

2. In Figure 5, while it was elegant to use Fos-based method (CANE) to selectively target 2MT activated PBel neurons, ChR2-stimulating axonal terminals of PBel in PSTh was significantly slower and less efficient at inducing skin/tail temperature changes compared to stimulating the cell bodies. The authors attributed this to low efficiency labeling using CANE or inability to activate sufficient number of axons. However, an alternative explanation is that PSTh is NOT the only target of PBel that are involved in 2MT-induced temperature change as it is known that PBel neurons project to multiple forebrain areas. It could be useful to show other axonal projection areas of the CANE-labeled PBel neurons, and then state the possibility that these other areas could also be involved.

RE: As suggested by Reviewer1, we found that CANE-labeled PBel neurons also projected into the central amygdala (CeA) and that optogenetic stimulation of these axon terminals in CeA induced a modest reduction in skin and core body temperature (Sup Fig 6). In the revision, we discussed that other projections areas of CANE-labeled PBel neurons might also be involved in 2MT-induced hypothermia.

3. After using the elegant CANE method to selectively label 2MT-activated PBel, the authors switched to the non-specific vGlut2-Cre to inhibit PB neurons using hM4Di in Figure 6. This is a questionable experiment. PB contains highly heterogeneous populations of neurons, including neurons activated by warm or cool body temperature (e.g. Geerling et al., Am. J. Physiol. Regul. Integr. Comp. Physiol., 2016). Not surprisingly, the results shown in Figure 6 is mild at best, likely due to activating neurons of opposing functions. If CANE is inefficient for performing inhibition experiments, the authors should at least use AAVretro-Flex-hMDi injected into PSTh and Cre virus injected into PBel, to express hM4Di-DREADDs in PBel neurons projecting to PSTh, and perform inhibition experiment.

RE: We thank Reviewer1 for the excellent suggestion. We performed the PBel-PSTh pathway-specific chemogenetic inhibition experiment by injecting AAV2/Retro-hSyn-Cre into the PSTh and AAV2/9-hSyn-DIO-hM4Di-mCherry (or control AAV2/9-hSyn-DIO-mCherry) into the PBel (new Fig 7). Accordingly, specific inhibition of PSTh-projecting PBel neurons significantly suppressed 2MT-induced tail temperature increase as well as skin and core body temperature reduction.

4. The authors suggest 2MT is sensed by TG or VG, and TG project to Sp5 and Sp5 project to PB. However, a recent paper (Rodriguez et al, Nature Neurosci. 2017) showed some TG neurons send monosynaptic projection directly to PB. The authors should revise their model to include this direct connection.

RE: We've included the direct projection from TG to PB in new Fig 8 and cited the reference.

Reviewer #2:

The authors have performed a series of retrograde tracing and optogenetic and dREADD experiments in TRPV1 KO and WT mice to discover the neural circuitry underlying the cutaneous vasodilatory and hypothermic response observed when mice are exposed to the scent of 2MT. The overall relevance to "fear" responses in animals other than mice or to stimuli other than 2MT, as well as several components of the proposed neural circuit for this hypothermia remain to be determined. The main novel insight provided by this study is the discovery that the PSTh contains neurons that can elicit a significant effect on T_{core} through inhibition of cutaneous vasoconstriction.

RE: We sincerely thank Reviewer2 for recognizing our new discovery.

Relevance: The authors attempt to couch the relevance of their study of this 2MT-evoked hypothermia/vasodilation in the framework of "innate fear", as opposed to "stress", which is accompanied by hyperthermia. However, they provide no teleological explanation or testable hypotheses for this thermoregulatory response to the odor of 2MT, nor are we provided with any examples in other species of a predator odor-driven hypothermia, or in general of a fight, flight or freeze situation that induces hypothermia. Overall, this anthropomorphic assignment of "fear" to this mouse vs predator encounter may not be valid, since no data are provided to show that a similar cutaneous vasodilation and hypothermia are evoked when a mouse is exposed to a predator whose odor is not an agonist of TRPA1. Thus, it might be more accurate to suggest that they are studying TRPA1-evoked hypothermia. To maintain this "fear" framework for their studies, the authors should demonstrate that other TRPA1 agonists, which are not predator odors, do not evoke a similar cutaneous vasodilation and hypothermia. It may simply be that this particular mouse predator odor happens to be a TRPA1 agonist.

RE: We thank Reviewer2 for raising an excellent point. Here we showed that cinnamaldehyde, a well-known TRPA1 agonist, could not cause hypothermia (Sup Fig 1a, 1b). This result further support our idea that 2MT induces acute hypothermia accompanying robust innate fear behaviors.

Nonetheless, these elegant experiments do reveal novel connections between the PBel, the PSTh and the NTS that contributes to TRPA1-evoked hypothermia and tail vasodilation in mice (the authors make a similar, albeit over-interpreted, conclusion in their Abstract). This is not a “circuit” since neither the sensory component (sensing the TRPA1 agonist) nor the primary efferent component (sympathetic premotor neurons for cutaneous vasoconstriction in the tail) are among these 3 synaptic sites, and the afferent and efferent connections to these 3 sites were not studied.

RE: We agree with the reviewer and change the word “circuit” to “pathway” in the revision.

*Approach: The powerful molecular/genetic techniques employed in this study significantly strengthen the authors’ conclusions regarding the connections among these 3 nuclei, as well as their functional connection to the 2MT stimulus. As the authors point out, the inability of the stimulation and inhibition experiments to completely mimic the magnitude of the original, 2MT stimulus or to completely block it could be the result of incomplete transduction or limited laser light exposure of relevant neurons – both of which are somewhat beyond the control of the investigators. Unfortunately, these ‘incomplete’ responses could also suggest that these 3 brain regions are not the only ones involved in the 2MT-evoked hypothermia. In this regard, the authors performed several anatomical (tracing, c-fos) experiments, but only report “selected” results. In the middle of page 6, for instance, the authors indicate that the PBel, PSTh and NTS had more c-fos+ neurons in *Trpa1*^{-/-} mice, but they do not provide a table of the c-fos+ neuron counts in the various brain nuclei examined in WT and KO mice, so we do not know how extensive their survey of c-fos+ neurons was, or whether other nuclei also had more c-fos+ neurons in the KO mice, suggesting a possible role in the hypothermic/vasodilatory response. In addition, the authors should mention the caveat that neurons that are inhibited during the 2MT response (these include at least the sympathetic premotor neurons controlling tail vasoconstriction) will not be c-fos+.*

RE: We thank Reviewer2 for positive comments and excellent suggestions. In our revision, we found the 2MT activated PBel neurons also projected to the CeA, and opto-activation of the axon terminals of 2MT activated PBel neurons in CeA could induce mild hypothermia (Sup Fig 6). Moreover, in a co-submitted manuscript, our collaborators Drs. Ko and Reiko Kobayakawa reported that trigeminal, vagal, or olfactory ablation could partially suppress 2MT-evoked hypothermia⁶². Therefore, we completely agree with Reviewer2 that multiple neural pathways are involved in the 2MT-evoked hypothermia and tail vasodilation responses.

As suggested by the reviewer, we performed comprehensive and comparative c-fos analysis of 2MT-treated WT and *Trpa1* KO mice (Fig. 1c, 1j and Sup Fig 1). In particular, we examined a dozen of known stress/fear, thermoregulation and vasodilation/vasoconstriction centers. Consistently, we found that WT brains showed significantly more c-fos positive neurons in the PBel, PSTh and NTS regions than *Trpa1* KO brains. By contrast, there was equivalent level of c-fos expression in the LPBD—a normal thermoregulatory center—between WT and KO brains. Interestingly, *Trpa1* KO brains relative to WT brains showed more c-fos expressing neurons in the POA area, such as MnPO (WT 287.29 c-fos+ neurons/mm², KO 385.88 c-fos+ neurons/mm²) and VMPO (WT 297.23 c-fos+ neurons/mm², KO 385.08 c-fos+ neurons/mm²). Because neurons that

are inhibited during 2MT response will not be c-fos+, this result suggests a possibility that some unknown neural pathway may contribute to 2MT-evoked hypothermia and tail vasodilation through suppression of MnPO/VMPO neurons. However, we observed that 2MT-activated PBel neurons did not project to the POA (Sup Fig 6a). suggesting that 2MT-activated PBel neurons represent a separate neuron population from those that respond to cold temperature and project to the POA (see below for detailed discussion of the vasomotor control centers, the rRpa, CVLM and RVLM).

Similarly, it is hard to interpret the data provided at the bottom of page 7 – what percent of c-fos+ neurons were also CTB+? This would indicate whether activated PSTh neurons might also project to other targets.

RE: We now quantified the percentage of c-fos+ neurons that were also CTB+ (Sup Fig 2 and 5). Our results showed that ~15% of c-fos+ neurons were CTB+ in PSTh (Sup Fig 2). This is probably due to limited labeling efficiency of CTB, but also indicates that there may be other projection targets for 2MT-activated PSTh neurons.

Again, on the top of page 12, the other projection sites of 2MT-activated PBel neurons should be reported. The data with dreadd inhibition of vGlut2+ neurons in PBel (page 13) are not interpretable, since this population of neurons is quite functionally diverse, with different projection targets, at least one of which (to the POA) plays a significant role in normal thermoregulatory responses.

Altogether, the authors' data only allow the conclusion that neurons connected among the 3 sites "contribute", or "are involved in" the 2MT-evoked response in mice. Considering the already demonstrated significant roles of the PBel and the NTS in thermoregulatory and BAT thermogenic control in rats, it seems difficult to believe that the neurons identified here in mice are not also involved in some aspect of normal thermoregulation or metabolic homeostasis, but the circuits for normal thermoregulatory responses have not been delineated in mice. It would have been very informative to confirm, for instance, that the PBel neurons identified with 2MT exposure are a separate population from those responding to cold exposure and projecting to the POA.

RE: We thank Reviewer2 for this excellent suggestion. We observed that CANE-labeled PBel neurons also projected into the CeA, and optogenetic stimulation of these axon terminals in CeA induced a modest reduction in skin and core body temperature (new Sup Fig 6). It has been reported that a group of PBel neurons respond to cold temperature and project to the preoptic area (POA)—a well-known thermoregulatory center. However, we found that 2MT-activated PBel neurons did not project to the POA (Sup Fig 6a). Moreover, there was slightly lower number of c-fos expressing neurons in the POA (e.g., MnPO and VMPO) of WT mice than *Trpa1* KO mice following 2MT treatment (Sup Fig 1c-11). Although we cannot completely rule out this possibility, it is less likely that the 2MT-activated PBel neurons project to POA to contribute to 2MT-evoked hypothermia. Finally, we performed chemogenetic inhibition of the PSTh-projecting PBel neurons by injecting AAV2/Retro-hSyn-Cre into the PSTh and AAV2/9-hSyn-DIO-hM4Di-mCherry (or AAV2/9-hSyn-DIO-mCherry) into PBel (new Fig 7). The PBel-PSTh specific inhibition significantly blunted 2MT-induced hypothermia and tail vasodilation (new Fig 7). Together, these

results strongly suggest that 2MT-activated PBel neurons represent a separate neural population from those responding to cold exposure and projecting to POA to regulate body temperature.

A shortcoming of this study is the absence of data on the activity of the mice during various manipulations. This is directly relevant since mice are highly dependent on activity thermogenesis to maintain their T_{core} in a subthermoneutral environment (testing in a thermoneutral experimental condition could have been informative in this study). If ‘freezing’ (i.e., no somatic movement) is maintained for several minutes, one would expect T_{core} to fall, independent of tail vasodilation or inhibition of BAT thermogenesis. Thus, if the 3 brain regions identified here are involved in inhibiting tail vasoconstriction, but not in inhibiting somatic movement, one might expect that activation of any one of these sites would completely recapitulate the cutaneous vasodilatory response, but not the hypothermia evoked by 2MT exposure. This was exactly the case with the TeLC silencing of PSTh neurons (page 9).

RE: We thank Reviewer2 for raising an excellent point. We went back to analyze the immobility rates of mice during all of the experiments in this study. However, we did not find any evidence supporting the idea that “freezing” could explain 2MT-induced hypothermia. For example, in the chemogenetic inhibition experiment of vGlut2+ PSTh neurons (top picture below), the mCherry-expressing and hM4Di-expressing mice showed similar immobility rate, but different T_{core} changes during 2MT exposure. In optogenetic stimulation experiment of the NTS-projecting PSTh neurons (bottom picture below), we again observed similar immobility rate, but very different T_{core} changes in EGFP-expressing and ChR2-expressing mice during photostimulation.

Telemetry recording of core temperature change and immobility rate.

(Immobility state: No displacement for 30 seconds or more)

Interpretation: The authors should discuss other functions of TRPA1-expressing afferents, since the life-long, global TRPA1 KO could affect a variety of other tonic and phasic reflexes involved in the homeostatic regulation of other, non-thermoregulatory systems (e.g., cardiovascular, respiratory) that might interfere or abrogate the responses to 2MT.

RE: Previous studies have shown that *Trpa1* KO mice exhibit no obvious developmental, physiological (e.g. cardiovascular, respiratory), or behavioral abnormalities except for insensitivity to nociceptive agents such as mustard oil (Bautista et al., 2006). However, it is still plausible that the lack of TRPA1 could somehow affect non-thermoregulatory systems under fearful conditions, which might interfere or abrogate the 2MT-evoked hypothermia.

The author should also indicate the proposed route for the 2MT molecules to gain access to the TRPA1 on the sensory terminals of trigeminal and vagal neurons. How have they ruled out the potential for these afferents to be providing a tonic input to the normal thermoregulatory pathways that is unrelated to sensing of 2MT, but is required for the elaboration of the 2MT cutaneous vasodilation?

RE: In this study, we showed convincing gain- and loss-of function experimental evidence to demonstrate that the PBel-PSTh-NTS pathway could play a crucial role in 2MT-evoked innate fear-associated hypothermia. However, we cannot rule out the possibility that 2MT-activated TG and vagal neurons could affect normal thermoregulatory pathways to contribute to 2MT-evoked hypothermia and tail vasodilation.

Curiously, the authors seem unaware of the significant literature on the CNS pathways specifically regulating cutaneous vasoconstriction! Due to its specific role in thermoregulation, blood flow in this vascular bed is controlled quite differently from that in other vascular beds. Rather than reference #46, the authors should study the work of Robin McAllen and of William Blessing, including their excellent review in Comprehensive Physiology. Sympathetic premotor neurons for cutaneous vasoconstriction are primarily contained in the medullary raphe pallidus nucleus, and it is the inhibition of these neurons that is likely involved in the cutaneous vasodilatory response to 2MT. Thus, the discussion of this issue on pages 16-17 should be completely rewritten.

Re: We are sorry for the omission and thank Reviewer2 for this excellent suggestion. In the revision, we cited the excellent work of Robin McAllen and William Blessing and discussed the possibility that inhibition of neurons in the medullary (rostral) raphe pallidus nucleus (rRpa) may contribute to the cutaneous vasodilatory response to 2MT. However, we observed equivalent level of c-fos expression in the rRpa of 2MT-exposed WT and *Trpa1* KO mice (Sup Fig 1). Thus, we think that the medullary raphe pallidus is not likely to play a critical role in 2MT-evoked tail vasodilation.

Rather, we propose that 2MT-activated NTS neurons may induce tail vasodilation by regulating the CVLM-RVLM vasomotor control pathway. We observed significant more c-fos expressing neurons in caudal ventrolateral medulla (CVLM) of WT mice than *Trpa1*^{-/-} mice (Supplementary Fig. 1l), and less c-fos expressing neurons in the rostral ventrolateral medulla (RVLM) in WT mice than *Trpa1*^{-/-} mice (Supplementary Fig. 1k). Excitation of the CVLM inhibitory neurons induces vasodilation through suppression of the neural activity in RVLM. Interestingly, CVLM receives direct synaptic input from NTS, which is strongly activated by 2MT exposure (Fig. 1i and 1j). These observations are consistent with our idea that 2MT-activated NTS neurons may induce tail vasodilation by regulating the CVLM-RVLM vasomotor control pathway.

Reviewer #3:

This is an interesting paper that demonstrates, and explores the regulation of, 2-methyl-2-thiazoline (2MT) induced hypothermia and elevated tail temperature in mice. A variety of methods are used including studies in mice lacking Trpa1, the chemosensor for 2MT, c-Fos to trace potential neural regulatory pathways, optogenetic activation and tetanus toxin light chain (TeLC)-mediated inactivation of neurons with identified areas. A pathway including the posterior subthalamic nucleus (PSTh), external lateral parabrachial subnucleus (PBel), and nucleus of the solitary tract (NTS) are implicated in regulating odor fear mediated hypothermia, and a proposed circuit is provided. The methods and analyses appear appropriate. A few suggestions are provided below.

RE: We thank Reviewer3 for the positive comments.

As written, the manuscript seems to imply that hypothermia is the more typical stress/fear response; however, hyperthermia is induced by multiple stressors, and fear, stress and emotion can produce the well-known stress-induced hyperthermia response. Thus, a discussion that compares different thermoregulatory responses and speculation as to how different fearful stimuli produce different thermal responses would be useful and very interesting. At the very least, a greater acknowledgement of the different responses should be provided. An example is that inescapable stress, mentioned in the Introduction and Discussion, needs to be specified because some types of inescapable stress, as well as escapable stress, produce hyperthermia. Restraint, which was cited as producing hypothermia, has also been reported to produce hyperthermia.

RE: We thank Reviewer3 for pointing this out. We re-wrote the Discussion to describe different types of stress/fear-associated thermoregulatory responses to highlight the fact that stress/fear-induced hyperthermia response is the more typical stress/fear response. In the Discussion, we speculate that extreme fear/stress stimuli induce hypothermia response to promote various bioprotective effects and the survival of the animals.

Seven days is a brief recovery period for telemetry implants. Is there a possible effect of recovery from surgery in the results? Also, how long were the mice allowed to recover from anesthesia prior to optogenetic stimulation?

RE: We agree with Reviewer3. We meant that our animals recovered for a minimum of seven days after surgery. To eliminate a possible effect of recovery from surgery in the results, we prepared control group of mice in all of the experiments, which were subjected to surgeries and recovery for the same period of time. We typically waited for ≥ 30 min to allow test mice to recover from anesthesia before optogenetic stimulation.

How was Adobe Photoshop used to analyze images?

RE: We used count tool of Adobe Photoshop and manually counted the number of labeled neurons in some histological experiments. We included the description in the Methods.

Preparation of the filter paper with TMT should be provided for both behavioral and c-fos studies.

RE: We dropped 20 ul of 2MT (2.1×10^{-4} mole) on the small piece of filter paper ($\sim 4 \text{ cm}^2$) and place on the center of temperature recording cage. We included the description in the Methods.

Overall, the paper reads well; however, it need significant editing with respect to grammar, particularly verb tense and the use of singular and plural nouns.

RE: We have corrected all typos and grammar mistakes in this revision.

Sincerely,

Qinghua Liu

Katsuyasu Sakurai

Reviewer #1 (Remarks to the Author):

The revised manuscript is significantly improved. The new experiments have largely addressed my previous concerns. The discovery that activation of PBel-PSTh-rostral NTS pathway control hypothermia associated with innate fear is novel and interesting.

I only have a few minor comments

(1) While I am glad to see the result that “cinnamaldehyde, a well-known Trpa1 agonist, did not induce hypothermia in wild-type mice”, in contrast to 2MT which also requires Trpa1 for sensing, the conclusion here should be: “Activation of Trpa1 is necessary but not sufficient to induce hypothermia”. It is likely, other sensory receptors in either somatosensory, vagal sensory, or olfactory sensory systems also detect 2MT, and the fear/hypothermia responses requires both Trpa1 and these other un-defined sensors.

(2) While I understand the rationale of the experiment of CTB-retrograde tracing from a defined nucleus (such as NTS) together with 2MT-induced Fos expression, to look for Fos+ neurons projecting to the starting nucleus. This rationale assumes “a DIRECT projection from Fos+ cells to the target nucleus is most relevant”. However, Fos+ neurons could be local interneurons that regulate activity of output neurons that then innervate the target nucleus, i.e. an INDIRECT pathway via local interneurons could be Equally Important.

I think the authors should make this clear, and the conclusions based on highest CTB/Fos overlap should be stated as implying a potential “direct projection from 2MT-activated neurons in xxx to xxx nucleus, although indirect pathways from Fos+ local interneurons to CTB-labeled projection neurons could also play important roles, we choose to focus on the direct pathway...”.

(3) The authors missed the recently published paper from Richard Palmiter’s lab in eLife (Bowen et al, 2020, <https://elifesciences.org/articles/59799>). In this eLife paper, they examined the functions of different axonal projection pathways from CGRP+ neurons in PBel to different downstream targets. Since PBel-CGRP is the major population of neurons in PBel, 2MT activated PBel neurons likely significantly overlap with PBel-CGRP neurons. Importantly, Bowen et al showed that activation of PBel-CGRP axons in PSTh, rostral CeA, and substantia innominate all induced reduction in skin temperature (Figure 3 in Bowen et al).

The authors should cite this paper and state that their results of activating PSTh and PBel-PSTh/PBel-CeA are consistent with Bowen et al. The authors in this manuscript did more careful analyses with both gain- and loss-of-function experiments, and identified the PSTh-rostral NTS pathway, so the novelty is NOT affected by Bowen et al., but that paper should be cited and discussed.

(4) In some experiments, the authors used AAV2/10-DIO-mCherry as control for AAV2/9-DIO-hM4Di. Please explain why two different serotypes are used, and whether serotype matters.

(5) Since many manipulations only resulted in partial effects on 2MT-induced hypothermia, the conclusion should be carefully drawn as “playing a major role” as opposed to “playing a critical role”.

(6) Most neurons in the brain have collaterals. Without seeing any data, it is difficult to imaging that PSTh neurons projecting to NTS (PSTh-NTS neurons) do NOT have any other targets, as the authors stated. Perhaps the authors could clarify this in their manuscript.

Reviewer #2 (Remarks to the Author):

The authors have made good progress in addressing previous reviewer concerns and in elaborating the details of the cutaneous vasodilation they observe following to exposure of mice to 2MT vapor. Although the data specifically related to the role of mouse tail vasodilation in the 2MT-evoked hypothermia and to the contribution of a PBel-PSTh-NTS neural pathway to this thermoregulatory response are solid, several aspects of the overall "fear odor" and TRPA1 framework for this study and the data interpretation require further consideration by the authors.

The authors provided new data indicating that similar exposure to the vapor of cinnamaldehyde, a non-2MT TRPA1 agonist, does not evoke the marked tail vasodilatory response elicited by exposure to 2MT. The absence of a hypothermic response to cinnamaldehyde, presumably a TRPA1 agonist equivalent to 2MT, contradicts the authors' implication that the TRPA1 channel is the "2MT receptor" that drives the hypothermic response. The authors suggest that this result indicates that the vasodilatory response to 2MT arises because 2MT is a component of "fear odor", whereas the absence of a similar response to cinnamaldehyde is because this TRPA1 agonist is not a component of a "fear odor". This line of reasoning is not logical – the TRPA1 channel does not "know" anything about an agonist molecule (e.g., part of an odor, etc) - the agonist either binds to the channel or it does not. The simplest explanation for the cinnamaldehyde result is that the 2MT interacts with a non-TRPA1 receptor (presumably in the olfactory bulb since it is a component of an "odor" and reaches the mouse in the form of a vapor) to activate the PBel-PSTh-NTS neural pathway. The cinnamaldehyde does not appear to be an agonist for this same non-TRPA1 receptor. The TRPA1 KO data indicate that TRPA1 channels on some population of neurons are required for the hypothermic response to 2MT. These TRPA1-expressing neurons may be located somewhere along the neural pathway beyond the olfactory neurons, or on neurons such as vagal or trigeminal afferents where such TRPA1 channels may be tonically activated to influence the ongoing level of cutaneous vasoconstriction, independently of 2MT exposure. In such a scenario, the TRPA1 channel could still be required for the hypothermic response, but the 2MT would not be activating the PBel-PSTh-NTS neural pathway by binding to this TRPA1 channel, otherwise the cinnamaldehyde should elicit the response as well. Of course, another explanation is that the requisite TRPA1 receptors are in the olfactory bulb and that the 2MT has access to these TRPA1 receptors, but somehow the cinnamaldehyde does not. Further experiments, such as c-fos and TRPA1 labeling in olfactory neurons after 2MT and after cinnamaldehyde exposures would address this critical question. Indeed, there may be other TRP channels, such as TRPV1 that also appears to play an ongoing role in normal thermoregulation (e.g., *J Neurosci.*, 31(5):1721-33, 2011), for which the KO mice would also fail to respond to 2MT. Clearly, the authors tested TRPA1 KO mice because 2MT is a TRPA1 agonist, but their cinnamaldehyde result is inconsistent with the TRPA1 channel being the "2MT receptor" that drives the observed hypothermia. It would be of interest to test the 2MT response in TRPV1 KO mice. Overall, this 2MT vs cinnamaldehyde conundrum and the TRPA1 receptor requirement necessitates further experimentation and/or significant manuscript editing by the authors. Since the authors' new data clearly indicates that the TRPA1 channel is not the olfactory "2MT receptor", the question arises as to the identity and location of the "2MT sensor" molecule. Within the authors' context of a "fear odor" and the methodological application of the 2MT as a vapor within the cage, the expectation is that the relevant 2MT binding molecule would be on the membrane of some population of olfactory neurons. In support of this hypothesis, the authors could test the thermoregulatory response of WT mice after temporarily blocking the nares. It would also

be relevant to determine if the fos+ neurons in the olfactory bulb express the TRPA1 channel. Overall, the framework for this paper, i.e., that the 2MT-evoked hypothermia represents a response to a “fear odor”, requires the authors to identify some important role for an olfactory stimulus in evoking the hypothermia observed with 2MT. Without this component, the very nice pathway work that is the main result of this manuscript loses its purported functional relevance. To maintain their framework, the authors could reference (or perform) experiments in which their temperature measurements were made when an actual predator (whose odor contains 2MT) was presented (through a protective screen) to the WT and KO mice. If none of these options are to be pursued, the fact that 2MT is a component of a known “fear odor” should be relegated to a paragraph in the Discussion. Indeed, there are a variety of other stimuli that elicit such a pronounced cutaneous vasodilation – have the authors observed fos+ cells within their PBel-PSTh-NTS neural pathway during exposure to a warm environment, during hypoglycemia, or during hypoxia?

The authors’ c-fos-based response to my earlier criticism of their suggestion that an NTS-CVLM-RVLM-sympathetic preganglionic neuron pathway underlies the 2MT-evoked cutaneous vasodilation is totally inadequate. The finding of an equivalent number of uncharacterized, c-fos+ neurons in the RPa in WT and KO mice provides no basis on which to rule out the critical role of the inhibition of the cutaneous vasoconstrictor sympathetic premotor neurons in the RPa in the tail vasodilatory response to 2MT. Similarly, the authors’ c-fos data for the RVLM and CVLM provides no basis on which to suggest that these brainstem regions play a role in the tail vasodilatory response to 2MT. To begin with, these c-fos studies comparing numbers of c-fos+ neurons between WT and KO after 2MT exposure are flawed because the authors did not perform the critical control experiment to determine the number of c-fos+ neurons in the WT and KO under the condition where they are exposed to a “sham” plectet of some volatile control substance (perhaps EtOH?). Then, the difference between the mean number of c-fos+ neurons after 2MT and after the sham exposure should be compared between WT and KO. This is the correct approach to such c-fos-based studies because it takes into account the level of c-fos generated by simply subjecting the mouse to the experimental conditions. In this case, for instance, the mouse is placed in a novel chamber to perform the temperature measurements during 2MT exposure, and both handling and a novel cage have been shown repeatedly to cause a stress-evoked activation of BAT which would involve activation of BAT sympathetic premotor neurons in RPa. Also related to methodology, such simple c-fos data cannot address the question of the brainstem circuitry mediating the hypothermic response because there is no functional, anatomical or phenotypic characterization of the neurons that would indicate whether they might be those expected to play a role in the hypothermic response. For instance, there were a huge (why so many?? A stress response in both WT and KO?) number of c-fos+ neurons in the RPa! But these could not be the glutamatergic, cutaneous vasoconstrictor sympathetic premotor neurons projecting to the spinal cord because these would be inhibited (i.e., no c-fos) during a vasodilatory response! Thus, the authors’ finding of a similar number of c-fos+ neurons in RPa is uninterpretable. Additionally, it is impossible to determine from Suppl Fig 1j at what rostral/caudal level of the RPa this analysis was performed – premotor neurons are located rostrally at the level of the facial nucleus. Similarly with the c-fos+ counts in CVLM and RVLM – because no control experiments were performed and no characterization of these neurons was obtained (the CVLM neurons the authors mention should project to the RVLM and be GABAergic), the c-fos data are uninterpretable.

Reviewer #3 (Remarks to the Author):

The authors have sufficiently addressed my concerns.

Point-to-point response letter to Reviewers' comments

Reviewer #1 (Remarks to the Author):

The revised manuscript is significantly improved. The new experiments have largely addressed my previous concerns. The discovery that activation of PBel-PSTh-rostral NTS pathway control hypothermia associated with innate fear is novel and interesting.

RE: We thank Reviewer 1 for recognizing the novelty and significance of our study.

I only have a few minor comments

(1) While I am glad to see the result that “cinnamaldehyde, a well-known Trpa1 agonist, did not induce hypothermia in wild-type mice”, in contrast to 2MT which also requires Trpa1 for sensing, the conclusion here should be: “Activation of Trpa1 is necessary but not sufficient to induce hypothermia”. It is likely, other sensory receptors in either somatosensory, vagal sensory, or olfactory sensory systems also detect 2MT, and the fear/hypothermia responses requires both Trpa1 and these other un-defined sensors.

RE: We agree with Reviewer 1 for this great suggestion. In our co-submitted manuscript (Matsuo et al., 2020, attached as related manuscript in this submission), our collaborators Drs. Kobayakawa performed a series of excellent experiments to clarify the discrepancy of why 2MT, but not cinnamaldehyde, induces hypothermic response. Remarkably, activation of TRPA1 by 2MT and cinnamaldehyde can induce different calcium influx kinetics, distinct gene expression profiles and cellular responses in TG neurons in vivo, which explains why 2MT, but not cinnamaldehyde, induces c-fos expression in downstream Sp5 neurons and the hypothermic response. In the revision, we concluded that TRPA1 was necessary but not sufficient to induce hypothermia, and described in detail Drs. Kobayakawa's results to further clarify this point in the Discussion (page 21, para 2).

(2) While I understand the rationale of the experiment of CTB-retrograde tracing from a defined nucleus (such as NTS) together with 2MT-induced Fos expression, to look for Fos+ neurons projecting to the starting nucleus. This rationale assumes “a DIRECT projection from Fos+ cells to the target nucleus is most relevant”. However, Fos+ neurons could be local interneurons that regulate activity of output neurons that then innervate the target nucleus, i.e. an INDIRECT pathway via local interneurons could be Equally Important.

I think the authors should make this clear, and the conclusions based on highest CTB/Fos overlap should be stated as implying a potential “direct projection from 2MT-activated neurons in xxx to xxx nucleus, although indirect pathways from Fos+ local interneurons to CTB-labeled projection neurons could also play important roles, we choose to focus on the direct pathway...”.

RE: We thank Reviewer 1 for the excellent suggestion. We clarified the direct or indirect pathways and stated carefully in both the Results and Discussion (page 8, para 2, line 11).

(3) The authors missed the recently published paper from Richard Palmiter's lab in eLife (Bowen et al, 2020, <https://elifesciences.org/articles/59799>). In this eLife paper, they examined the functions of different axonal projection pathways from CGRP+ neurons in PBel to different downstream targets. Since PBel-CGRP is the major population of neurons in PBel, 2MT activated PBel neurons likely significantly overlap with PBel-CGRP neurons.

Importantly, Bowen et al showed that activation of PBel-CGRP axons in PSTh, rostral CeA, and substantia innominate all induced reduction in skin temperature (Figure 3 in Bowen et al).

The authors should cite this paper and state that their results of activating PSTh and PBel-PSTh/PBel-CeA are consistent with Bowen et al. The authors in this manuscript did more careful analyses with both gain- and loss-of-function experiments, and identified the PSTh-rostral NTS pathway, so the novelty is NOT affected by Bowen et al., but that paper should be cited and discussed.

RE: We are sorry for the omission and thank Reviewer 1 for this excellent suggestion. In the revision, we cited and discussed the excellent work of Richard Palmiter (page 7, para 2, line 6; page 15, para 3, line 8).

(4) In some experiments, the authors used AAV2/10-DIO-mCherry as control for AAV2/9-DIO-hM4Di. Please explain why two different serotypes are used, and whether serotype matters.

RE: Both serotypes work well in the CNS. We have no special reason to use different serotypes.

(5) Since many manipulations only resulted in partial effects on 2MT-induced hypothermia, the conclusion should be carefully drawn as “playing a major role” as opposed to “playing a critical role”.

RE: We thank Reviewer 1 for this great suggestion. In the revision, we described as “playing a major role” instead of “playing a critical role”.

(6) Most neurons in the brain have collaterals. Without seeing any data, it is difficult to imagine that PSTh neurons projecting to NTS (PSTh-NTS neurons) do NOT have any other targets, as the authors stated. Perhaps the authors could clarify this in their manuscript.

RE: We agree with Reviewer 1 and clarified this point in the revised manuscript (page 11, para 1, line 3).

Reviewer #2 (Remarks to the Author):

The authors have made good progress in addressing previous reviewer concerns and in elaborating the details of the cutaneous vasodilation they observe following to exposure of mice to 2MT vapor. Although the data specifically related to the role of mouse tail vasodilation in the 2MT-evoked hypothermia and to the contribution of a PBel-PSTh-NTS neural pathway to this thermoregulatory response are solid, several aspects of the overall "fear odor" and TRPA1 framework for this study and the data interpretation require further consideration by the authors.

RE: We thank Reviewer 2 for recognizing that we “have made good progress in addressing his previous concerns...the contribution of a PBel-PSTh-NTS neural pathway to this thermoregulatory response are solid”. It is important to emphasize that our manuscript focus on the discovery of a novel PBel-PSTh-NTS neural pathway that underlies 2MT-evoked hypothermia. We simply place our finding in the context of a large body of previous and current experimental results to make sense of the molecular/neural mechanisms and biological purposes of these interesting phenomena (page 4, para 2; page 19, para 1, line 7).

Notably, the “fear odor” and TRPA1 framework has been well established by our previous publication (Wang et al., Nature Communications 2018), in which we identified TRPA1 as a novel chemosensor for 2MT/TMT (fox odor) and snake skin-evoked innate fear behaviors, through unbiased forward genetic screening of ~14,000 randomly mutagenized mice. Furthermore, our collaborators, Dr.

Reiko and Ko Kobayakawa, have observed that innate fear odor 2MT elicits physiological responses, such as bradycardia, hypothermia and hypometabolism, that provide strong bioprotective effects against hypoxia and ischemia/reperfusion injuries (Isosaka et al., Cell 2015; Matsuo et al., Commun Biol 2021). Along with our manuscript, Drs. Kobayakawa also co-submitted a manuscript titled "**TRPA1 commands intrinsic bioprotective effects of innate fear odors**" (Matsuo et al., 2020, attached as related manuscript in this submission), which is now accepted in principle by Nature Communications.

We summarized below the most significant and relevant results that establish the large framework of “fear odor” and “TRPA1 as 2MT receptor” from these previous and current studies as described above.

Figure 1. TrpA1 functions as a specific chemosensor for 2MT/TMT.

- 1) *Trpa1* KO mice are defective for 2MT/TMT/snake skin-induced innate fear behaviors (e.g., freezing avoidance, risk assessment) and physiological responses (e.g., hypothermia, bradycardia, and stress hormone surge);
- 2) *Trpv1* KO mice show normal 2MT/TMT-evoked freezing and 2MT-induced hypothermic response, although ~90% of TRPA1-expressing neurons also express TRPV1 (page 4, para 2).
- 3) Habituation-dishabituation test shows that WT and *Trpa1* KO mice show equal sensitivity of smell of 2MT; WT and *Trpa1* KO littermates show similar 2MT-evoked c-fos expression in the olfactory system (Wang et al., Nature Communications 2018);
- 4) TRPA1 functions as a highly specific sensor for 2MT/TMT among the TRP family of proteins, as shown by calcium imaging in transfected HEK293 cells (Figure 1A and B) (Wang et al., Nature Communications 2018);
- 5) 2MT/TMT activates TRPA1 through covalent modification of key cysteine residues of TRPA1, as shown by site-directed mutagenesis and click chemistry experiments (Figure 1C) (Wang et al., Nature Communications 2018);
- 6) TRPA1 is essential for 2MT sensing by a subset of trigeminal ganglion (TG) neurons (Figure 1D) (Wang et al., Nature Communications 2018);
- 7) TRPA1⁺ TG neurons contribute critically to 2MT-evoked innate freezing, as shown by unilateral TG lesion experiment in WT mice and TG-specific AAV-TRPA1 rescue experiment in *Trpa1* KO mice;

(Wang et al., Nature Communications 2018);

- 8) Chemogenetic activation of Sp5-projecting TRPA1⁺ TG neurons is sufficient to induce hypothermia (Drs. Kobayakawa's co-submitted a manuscript);
- 9) 2MT-evoked hypothermia offer bioprotective effects against hypoxia-ischemic injuries in mice (Matsuo et al., Commun Biol 2021);

Taken together, these comprehensive and consistent experimental results demonstrate that TRPA1 is the principal chemosensor for 2MT-evoked innate fear/defensive behaviors (e.g., freezing) & physiological responses (e.g., hypothermia) to promote survival in life-threatening conditions.

The authors provided new data indicating that similar exposure to the vapor of cinnamaldehyde, a non-2MT TRPA1 agonist, does not evoke the marked tail vasodilatory response elicited by exposure to 2MT. The absence of a hypothermic response to cinnamaldehyde, presumably a TRPA1 agonist equivalent to 2MT, contradicts the authors' implication that the TRPA1 channel is the "2MT receptor" that drives the hypothermic response. The authors suggest that this result indicates that the vasodilatory response to 2MT arises because 2MT is a component of "fear odor", whereas the absence of a similar response to cinnamaldehyde is because this TRPA1 agonist is not a component of a "fear odor". This line of reasoning is not logical – the TRPA1 channel does not "know" anything about an agonist molecule (e.g., part of an odor, etc) - the agonist either binds to the channel or it does not. The simplest explanation for the cinnamaldehyde result is that the 2MT interacts with a non-TRPA1 receptor (presumably in the olfactory bulb since it is a component of an "odor" and reaches the mouse in the form of a vapor) to activate the PBel-PSTh-NTS neural pathway. The cinnamaldehyde does not appear to be an agonist for this same non-TRPA1 receptor.

RE: We thank Reviewer 2 for stimulating discussion and alternative hypothesis on the different hypothermic response to 2MT vs. cinnamaldehyde. However, we respectfully disagree with Reviewer 2's reasoning and conclusions. It is worth noting that Reviewer 2's hidden assumption was that all TRPA1 agonists must induce hypothermia, otherwise, TRPA1 should not be the receptor for 2MT-evoked hypothermia.

Rather, we believe that biological responses are often not a simple binary response, and a negative result usually has multiple alternative interpretations. For example, 1) 2MT and CNA may have different affinity for TRPA1; 2) 2MT and CNA may activate TRPA1 with distinct mechanisms or kinetics; 3) 2MT and CNA may cause qualitatively different cellular responses to 2MT in TRPA1⁺ neurons.

In our co-submitted manuscript, Drs. Kobayakawa performed a series of excellent experiments to clarify the discrepancy of why 2MT, but not CNA, induces hypothermia in mice.

- 1) In vivo calcium imaging indicates that 2MT and CNA evoke different calcium influx kinetics in TG neurons (Figure 2A). We previously showed that TRPA1 is essential for 2MT sensing by TG neurons;
- 2) RNA-seq experiments reveal that 2MT and CNA induce distinct genes expression profiles in the TG (Figure 2B), indicative of qualitatively different cellular responses in TG neurons;
- 3) 2MT, but not CNA, can induce c-fos expression in Sp5 neurons (Figure 2C), which receives direct input from TG neurons. We previously showed TRPA1 is essential for 2MT-induced c-fos expression in Sp5;
- 4) Chemogenetic activation of the Sp5-projecting TRPA1⁺ TG neurons is sufficient to induce hypothermia (Figure 2D), suggesting that TRPA1⁺ TG neurons contribute critically to 2MT-evoked hypothermia;
- 5) Although ~90% of TRPA1⁺ neurons also express TRPV1, *Trpv1* KO mice show normal 2MT-evoked hypothermic response, suggesting TRPV1 is not the receptor for 2MT (Figure 2E).

Figure 2. 2MT and CNA induce different biological responses at multiple levels.

These results suggest that, remarkably, activation of TRPA1 by 2MT and CNA induces distinct kinetics of calcium influx and gene expression profiles in TRPA1⁺ TG neurons, which may explain the discrepancy in the information transmitted to downstream Sp5 neurons to trigger hypothermia. Although these lines of evidence could explain the different responses to 2MT and CNA, we expect that readers might have a similar question as Reviewer 2. Thus, in the revision, we concluded that activation of TRPA1 was necessary, but not sufficient to induce hypothermia, and described in detail Drs. Kobayakawa's results to further clarify this point in the Discussion (page 21, para 2).

The TRPA1 KO data indicate that TRPA1 channels on some population of neurons are required for the hypothermic response to 2MT. These TRPA1-expressing neurons may be located somewhere along the neural pathway beyond the olfactory neurons, or on neurons such as vagal or trigeminal afferents where such TRPA1 channels may be tonically activated to influence the ongoing level of cutaneous vasoconstriction, independently of 2MT exposure. In such a scenario, the TRPA1 channel could still be required for the hypothermic response, but the 2MT would not be activating the PBel-PSTh-NTS neural pathway by binding to this TRPA1 channel, otherwise the cinnamaldehyde should elicit the response as well. Of course, another explanation is that the requisite TRPA1 receptors are in the olfactory bulb and that the 2MT has access to these TRPA1 receptors, but somehow the cinnamaldehyde does not. Further experiments, such as c-fos and TRPA1 labeling in olfactory neurons after 2MT and after cinnamaldehyde exposures would address this critical question.

RE: We thank Reviewer 2 for interesting suggestions. In their co-submitted manuscript (Matsuo et al., 2020, attached as related manuscript in this submission), Drs. Kobayakawa showed several important results suggesting that the trigeminal, vagus, and olfactory systems are involved in the 2MT-evoked hypothermic response as shown below.

Figure 3. Multiple sensory pathways are involved in 2MT-evoked hypothermia.

- 1) Olfactory bulbectomized mice show partial suppression of 2MT-induced hypothermia, and ablation of dorsal zone olfactory sensory neurons also attenuates 2MT-induced hypothermia (Figure 3A);
- 2) Ablation of bilateral vagus nerves below diaphragm attenuates 2MT-induced hypothermia (Figure 3B);
- 3) Unilateral TG lesion significantly blunts 2MT-evoked hypothermia (Figure 3C); bilateral TG lesion causes lethality for unknown reasons;
- 4) Olfactory-specific *Trpa1* KO mice (crossing OMP-Cre and *Trpa1* floxed mice) did not affect 2MT-induced hypothermia, suggesting contribution of olfactory system is independent of TRPA1 (Figure 3D);
- 5) TG-specific *Trpa1* KO mice (crossing Advillin-Cre mice and *Trpa1* floxed mice) blunts 2MT-induced hypothermia, suggesting the contribution of TG neurons is TRPA1-dependent (Figure 3E);

Taken together, Drs. Kobayakawa and our studies strongly suggest that the PBel-PTSh-NTS pathway functions downstream of TRPA1⁺ TG and VG neurons. Accordingly, TG neurons project to Sp5, both of which project to PBel, whereas VG neurons project to NTS. The lack of TRPA1 diminishes 2MT-induced c-fos expression in all of these brain areas, and abolishes the 2MT-induced hypothermia and tail vasodilation. These striking phenotypes of *Trpa1* KO mice is in big contrast to the partial attenuation effects of 2MT-evoked hypothermia as a result of ablation of the olfactory, vagus, or trigeminal pathway. These results strongly argue that TRPA1 is the principal chemosensor that mediates 2MT-induced hypothermia. However, we do not exclude the possibility that other receptors may also contribute to this response. In the revision, we described Drs. Kobayakawa's results supporting the involvement of multiple neural pathways in 2MT-induced hypothermia (page 18, para 2, line 12).

Indeed, there may be other TRP channels, such as TRPV1 that also appears to play an ongoing role in normal thermoregulation (e.g., *J Neurosci.*, 31(5):1721-33, 2011), for which the KO mice would also fail to respond to 2MT. Clearly, the authors tested TRPA1 KO mice because 2MT is a TRPA1 agonist, but their cinnamaldehyde result is inconsistent with the TRPA1 channel being the "2MT receptor" that drives the observed hypothermia. It would be of

interest to test the 2MT response in TRPV1 KO mice. Overall, this 2MT vs cinnamaldehyde conundrum and the TRPA1 receptor requirement necessitates further experimentation and/or significant manuscript editing by the authors. RE: We thank Reviewer2 for this excellent suggestion. Although the majority (>90%) of TRPA1⁺ neurons also express TRPV1, *Trpv1* KO mice show normal 2MT-evoked hypothermia, suggesting that TRPV1 is not the receptor for 2MT (Figure 2E). Moreover, we have previously shown that TRPA1 functions as a highly specific sensor for 2MT/TMT (fox odor) among the TRP family of proteins, as shown by calcium imaging in transfected HEK293 cells (Figure 1A and B) (Wang et al., Nature Communications 2018).

Since the authors' new data clearly indicates that the TRPA1 channel is not the olfactory "2MT receptor", the question arises as to the identity and location of the "2MT sensor" molecule. Within the authors' context of a "fear odor" and the methodological application of the 2MT as a vapor within the cage, the expectation is that the relevant 2MT binding molecule would be on the membrane of some population of olfactory neurons. In support of this hypothesis, the authors could test the thermoregulatory response of WT mice after temporarily blocking the nares. It would also be relevant to determine if the fos⁺ neurons in the olfactory bulb express the TRPA1 channel. Overall, the framework for this paper, i.e., that the 2MT-evoked hypothermia represents a response to a "fear odor", requires the authors to identify some important role for an olfactory stimulus in evoking the hypothermia observed with 2MT. Without this component, the very nice pathway work that is the main result of this manuscript loses its purported functional relevance. To maintain their framework, the authors could reference (or perform) experiments in which their temperature measurements were made when an actual predator (whose odor contains 2MT) was presented (through a protective screen) to the WT and KO mice. If none of these options are to be pursued, the fact that 2MT is a component of a known "fear odor" should be relegated to a paragraph in the Discussion.

RE: As discussed above, we respectfully disagree with Reviewer 2's conclusion that our new data clearly indicates that TRPA1 is not the "2MT receptor", from which most of Reviewer 2's arguments originated.

In our co-submitted manuscript, Drs. Kobayakawa found that olfactory bulbectomized mice show slight attenuation of 2MT-induced hypothermia (Figure 3A), indicative of at most a minor contribution from the olfactory system. Moreover, olfactory-specific *Trpa1* KO mice (by crossing OMP-Cre and *Trpa1* floxed mice) did not affect 2MT-induced hypothermia, suggesting the minor contribution of olfactory system is independent of TRPA1 (Figure 3D);

In our previous publication (Wang et al., Nature Communications 2018), we reported that 2MT could induce equivalent levels of c-fos expression in the olfactory bulb and olfactory cortex. Habituation-dishabituation assays indicates that WT and *Trpa1* KO mice show equal sensitivity of smell of 2MT. Despite that TRPA1 is dispensable for "smelling" 2MT by the olfactory system, *Trpa1* KO mice is defective for 2MT-evoked innate fear behaviors and physiological responses. Thus, TRPA1-expressing somatosensory (e.g. trigeminal and vagus) system, rather than the olfactory system, is primarily responsible for 2MT-evoked innate fear behaviors and physiological responses.

It is important to emphasize that our manuscript is all about the discovery of a novel PBel-PSTh-NTS neural pathway that underlies 2MT-evoked hypothermia. In the Introduction and Discussion, we simply place our findings in the context of a large body of previous and current experimental results to make sense of the molecular/neural mechanisms and biological purposes of these interesting phenomena.

Indeed, there are a variety of other stimuli that elicit such a pronounced cutaneous vasodilation – have the authors observed fos⁺ cells within their PBel-PSTh-NTS neural pathway during exposure to a warm environment, during hypoglycemia, or during hypoxia?

RE: While it is interesting to investigate whether the PBel-PSTh-NTS pathway is also involved in the cutaneous vasodilation response to warm environment, hypoglycemia, or hypoxia, we believe that these experiments are beyond the scope of our current study.

The authors' c-fos-based response to my earlier criticism of their suggestion that an NTS-CVLM-RVLM-sympathetic preganglionic neuron pathway underlies the 2MT-evoked cutaneous vasodilation is totally inadequate. The finding of an equivalent number of uncharacterized, c-fos⁺ neurons in the RPa in WT and KO mice provides no basis on which to rule out the critical role of the inhibition of the cutaneous vasoconstrictor sympathetic premotor neurons in the RPa in the tail vasodilatory response to 2MT. Similarly, the authors' c-fos data for the RVLM and CVLM provides no basis on which to suggest that these brainstem regions play a role in the tail vasodilatory response to 2MT. To begin with, these c-fos studies comparing numbers of c-fos⁺ neurons between WT and KO after 2MT exposure are flawed because the authors did not perform the critical control experiment to determine the number of c-fos⁺ neurons in the WT and KO under the condition where they are exposed to a “sham” plectet of some volatile control substance (perhaps EtOH?). Then, the difference between the mean number of c-fos⁺ neurons after 2MT and after the sham exposure should be compared between WT and KO. This is the correct approach to such c-fos-based studies because it takes into account the level of c-fos generated by simply subjecting the mouse to the experimental conditions. In this case, for instance, the mouse is placed in a novel chamber to perform the temperature measurements during 2MT exposure, and both handling and a novel cage have been shown repeatedly to cause a stress-evoked activation of BAT which would involve activation of BAT sympathetic premotor neurons in RPa. Also related to methodology, such simple c-fos data cannot address the question of the brainstem circuitry mediating the hypothermic response because there is no functional, anatomical or phenotypic characterization of the neurons that would indicate whether they might be those expected to play a role in the hypothermic response. For instance, there were a huge (why so many?? A stress response in both WT and KO?) number of c-fos⁺ neurons in the RPa! But these could not be the glutamatergic, cutaneous vasoconstrictor sympathetic premotor neurons projecting to the spinal cord because these would be inhibited (i.e., no c-fos) during a vasodilatory response! Thus, the authors' finding of a similar number of c-fos⁺ neurons in RPa is uninterpretable. Additionally, it is impossible to determine from Suppl Fig 1j at what rostral/caudal level of the RPa this analysis was performed – premotor neurons are located rostrally at the level of the facial nucleus. Similarly with the c-fos⁺ counts in CVLM and RVLM – because no control experiments were performed and no characterization of these neurons was obtained (the CVLM neurons the authors mention should project to the RVLM and be GABAergic), the c-fos data are uninterpretable.

RE: We thank Reviewer 2 for raising an excellent point. As suggested by Reviewer 2, we performed comparative analysis of c-fos expression among saline or 2MT-treated WT mice and 2MT-treated *Trpa1* KO mice. 2MT exposure specifically induced c-fos expression in the rRpa, RVLM and CVLM of WT mice (Figure 4, n=4). In *Trpa1* KO mice relative to WT mice, RVLM showed increased 2MT-induced c-fos expression, whereas CVLM showed decreased 2MT-induced c-fos expression. By contrast, rRpa showed no significant difference in 2MT-induced c-fos expression between WT and *Trpa1* KO mice.

Figure 4. c-fos expression of RVLM, CVLM and rRpa.

We totally agree with Reviewer 2 that it would take significant time and efforts to determine precisely which neural pathway mediates the tail vasodilation in response to 2MT, which is beyond the scope of the current study and probably another project on its own. Thus, we believe that the most sensible way forward is to remove the c-fos staining data of rRPa, CVLM and RVLM from Sup Fig 1 and simply discuss potential neural pathways that may mediate 2MT-induced tail vasodilation (page 20, para 3; page 21, para 1). Future studies are urgently needed to investigate precisely which of these brainstem nuclei play a major role in 2MT-induced tail vasodilation downstream of the PBel-PSTh-NTS pathway.

Reviewer #3 (Remarks to the Author):

The authors have sufficiently addressed my concerns.

RE: We thank Reviewer 3 for approving our revision.

Reviewer #1 (Remarks to the Author):

The authors have fully addressed all my concerns in this re-revised manuscript. I support the publication of this manuscript in Nature Communications.

Reviewer #2 (Remarks to the Author):

This is indeed a complex afferent system, especially when a substance that is continuously described as an "odor" turns out to be sensed by TG and VG neurons, rather than via the olfactory system! One can imagine the requisite sensing receptors on the cutaneous processes of TG neurons, but how the "odor" gets to VG neuronal processes is puzzling...perhaps those VG afferent in the lower airways. The companion paper clarifies many of the issues that seemed problematic in interpreting the original submission - it is unfortunate that this manuscript was not made available with the earlier submission.